# Confusion-Driven Self-Supervised Progressively Weighted Ensemble Learning for Non-Exemplar Class Incremental Learning

**Kai Hu**
Xiangtan University
kaihu@xtu.edu.cn

**Zhang Yu**
Xiangtan University
202221633002@smail.xtu.edu.cn

**Yuan Zhang**[*]
Xiangtan University
yuanz@xtu.edu.cn

**Zhineng Chen**
Fudan University
zhinchen@fudan.edu.cn

**Xieping Gao**
Hunan Normal University
xpgao@hunnu.edu.cn

## Abstract

Non-exemplar class incremental learning (NECIL) aims to continuously assimilate new knowledge while retaining previously acquired knowledge in scenarios where prior examples are unavailable. A prevalent strategy within NECIL mitigates knowledge forgetting by freezing the feature extractor after training on the initial task. However, this freezing mechanism does not provide explicit training to differentiate between new and old classes, resulting in overlapping feature representations. To address this challenge, we propose a **C**onfusion-driven se**L**f-supervised pr**O**gressi**V**ely weighted **E**nsemble lea**R**ning (*CLOVER*) framework for NECIL. Firstly, we introduce a confusion-driven self-supervised learning approach that enhances representation extraction by guiding the model to distinguish between highly confusable classes, thereby reducing class representation overlap. Secondly, we develop a progressively weighted ensemble learning method that gradually adjusts weights to integrate diverse knowledge more effectively, further minimizing representation overlap. Finally, extensive experiments demonstrate that our proposed method achieves state-of-the-art results on the CIFAR100, TinyImageNet, and ImageNet-Subset NECIL benchmarks. [2]

## 1 Introduction

Over the past few decades, deep neural networks have exhibited remarkable performance in various domains [1, 2, 3, 4, 5]. Given that the real world is both open and dynamic, it is crucial for models to continuously acquire and integrate new knowledge. However, directly fine-tuning a model on new data often results in catastrophic forgetting [6], where the performance on previously encountered data deteriorates significantly. To address this challenge, several class incremental learning (CIL) methods [7, 8] store a subset of previous exemplars and replay them during the learning of new tasks to preserve prior knowledge. Nevertheless, due to privacy concerns and storage limitations, non-exemplar class incremental learning (NECIL) [9], which aims to continuously adapt to new classes while preventing forgetting previously learned ones without retaining earlier samples, has gained increasing attention from researchers.

Existing NECIL methods can be broadly classified into two categories: prototype-based methods and frozen feature extractor-based methods, based on their training paradigms. Prototype-based

---

[*]Corresponding author.
[2]Code is publicly available at: https://github.com/MLMIP/CLOVER

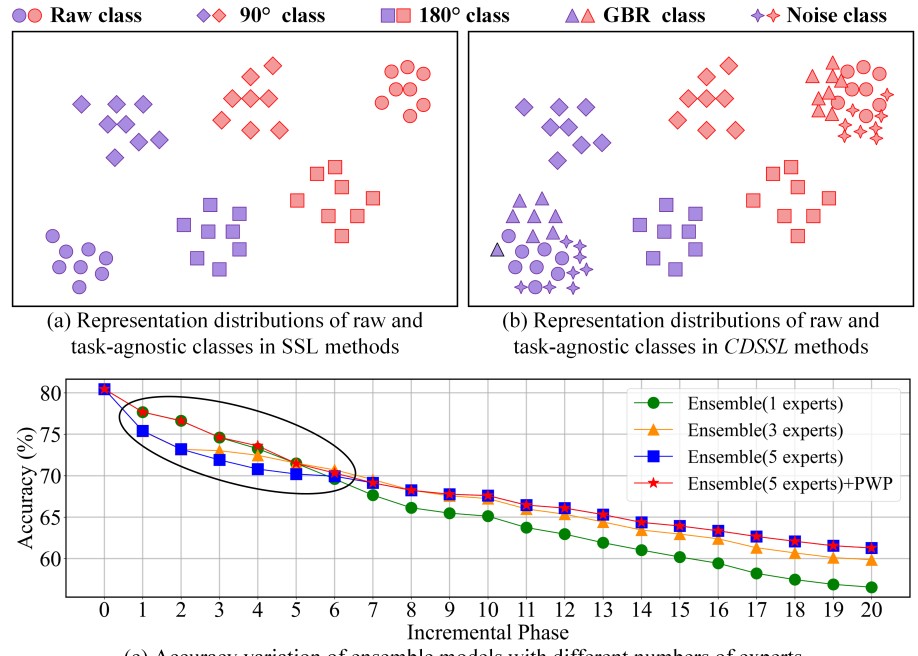

(a) Representation distributions of raw and task-agnostic classes in SSL methods

(b) Representation distributions of raw and task-agnostic classes in *CDSSL* methods

(c) Accuracy variation of ensemble models with different numbers of experts

Figure 1: (a) In self-supervised learning, substantial discrepancies exist in the representation distributions between raw classes and task-agnostic classes (sharing the same color but differing in shape). Training a model to differentiate between these classes does not effectively enhance its capability for representation extraction. (b) *CDSSL* introduces highly confusable task-agnostic classes, with GBR and noise classes generated through color channel swapping and noise injection, respectively. This approach aims to encourage the model to extract more discriminative representations. (c) The accuracy curves for the ensemble model with varying numbers of experts indicate that adding a new expert (during phases 1 and 3) consistently leads to performance degradation, which is effectively mitigated by the proposed *PWP* strategy.

methods [9, 10, 11, 12] take advantage of the mean of class representations as prototypes to maintain decision boundaries during model fine-tuning for incremental tasks. These prototypes serve as substitutes for samples from previous classes, conveying distributional information that effectively mitigates catastrophic forgetting. However, as the model undergoes continuous updates, these approaches inevitably experience prototype degradation [12], wherein the preserved prototypes of old classes diverge from their true distributions. In contrast, frozen feature extractor-based methods [13, 14, 15] avoid degradation by freezing the feature extractor after training on the initial task, while maintaining the stability of representations and mitigating forgetting. Despite these advantages, a notable drawback of this method is its propensity to induce representation overlap between new and old classes. As the model is optimized solely for the initial class distribution, it lacks the necessary adaptability to incorporate new class data effectively. Consequently, representations of new classes often exhibit dispersed distributions or overlap with those of existing classes within a fixed representation space, particularly when similarities exist along certain dimensions. This phenomenon hinders the establishment of clear decision boundaries and significantly impairs classification performance in incremental learning tasks.

There are currently two primary approaches to addressing representation overlap: self-supervised learning (SSL) and ensemble learning. SSL [9, 16, 17] is utilized in the initial task to train the feature extractor. SSL applies transformations to raw classes and assigns novel task-agnostic labels, thereby constructing task-agnostic classes. Subsequently, by encouraging the model to discriminate among these classes, SSL enhances the model's generalization capability and effectively mitigates representation overlap and confusion [18]. However, most studies [9, 18, 19, 20, 21] integrate SSL with additional techniques aimed at improving representation extraction, thereby leaving much of SSL's inherent potential largely untapped. As shown in Fig. 1(a), the rotation transformation employed in SSL significantly alters the representational distribution of raw classes, enabling the model to distinguish these classes more easily even without SSL. As a result, it fails to effectively enhance

the model's ability to extract discriminative representations. Conversely, ensemble learning [22, 23, 24] capitalizes on multiple expert models to extract discriminative representations for each class from diverse perspectives. This method effectively differentiates between new and old class representations while minimizing their overlap. Nonetheless, introducing a new expert can lead to performance degradation (see Fig. 1(c)). This issue stems from substantial heterogeneity in the predictive capabilities exhibited by the experts. The first expert, trained on the base task, exhibits stronger predictive capability and covers a broader range of classes than experts trained on incremental tasks. This makes the first expert more reliable in the early training stages, while later experts remain unreliable for a period after their introduction. Thus, a newly added expert, often with relatively limited predictive capability, can interfere with predictions made by more experienced experts.

To address the aforementioned challenges, we introduce a novel NECIL framework, named **C**onfusion-driven se**L**f-supervised pr**O**gressi**V**ely weighted **E**nsemble lea**R**ning (*CLOVER*). In this framework, we propose a Confusion-Driven Self-Supervised Learning (*CDSSL*) method that enhances the model's capability for representation extraction by guiding it to differentiate between highly confusing classes generated through color channel swapping and noise inception transformations (see Fig. 1(b)). This approach effectively alleviates the issue of representation overlap between new and old classes. Furthermore, we introduce a Progressively Weighted Prediction (*PWP*) strategy aimed at mitigating the influence of unreliable experts within the ensemble model. Specifically, we assign an initially low weight to new experts and incrementally increase their weight task by task until it reaches a predetermined maximum value. The final prediction is then computed as a weighted sum of the predictions from all experts. In conclusion, our main contributions are summarized as follows:

- We propose a **C**onfusion-driven se**L**f-supervised progressi**V**ely weighted **E**nsemble lea**R**ning framework named *CLOVER* for NECIL. This framework improves the model's capability for representation extraction by learning task-agnostic classes and diverse ensemble predictions. These advancements strengthen the discriminative power for new classes, thereby mitigating the representation overlap with old classes.

- We propose a confusion-driven self-supervised learning method that enhances the ability to extract discriminative representations by guiding it to differentiate between highly confusing task-agnostic classes generated through color channel swapping and noise injection. Furthermore, we introduce a progressively weighted prediction strategy that gradually increases the weight of new experts to mitigate performance degradation caused by unreliable new experts within the ensemble model.

- Extensive experimental results on CIFAR100, TinyImageNet and ImageNet-Subset have demonstrated the superior performance of our proposed *CLOVER* over other NECIL approaches.

## 2 Related Work

### 2.1 Non-Exemplar Class Incremental Learning

In general, NECIL methods can be broadly categorized into prototype-based methods [9, 12] and frozen feature extractor-based methods [14, 15, 25].

Prototype-based methods investigate strategies for leveraging prototypes to deliver more accurate insights regarding prior tasks, thereby effectively mitigating the phenomenon of knowledge forgetting. For example, PASS [9] employed Gaussian noise to augment prototypes, providing distributional information of old classes within the representation space to maintain decision boundaries of previous tasks. SSRE [10] identified the primary challenge in NECIL as the overlap between the representations of new and old classes and addressed this issue through a prototype selection mechanism, which used prototypes to preserve prior knowledge while simultaneously facilitating the learning of new tasks. NAPA-VQ [11] learned the topological relationships of the feature space to identify confusable neighboring classes and used this information to generate representative prototypes for old classes, thereby facilitating the delineation of boundaries between new and old classes and reducing overlap. However, prototype-based methods inevitably suffer from representation shifts caused by the continual updates to the feature extractor. To address this issue, a variety of methods [12, 26, 27] have been proposed to alleviate representation shift by reconstructing the prototypes of previously learned classes, thereby enhancing the accuracy and effectiveness of the information they convey about the old

class distributions. More recently, DCMI [28] proposed a novel generation method for synthesizing images with both semantic and domain consistency, which effectively facilitated knowledge retention.

On the other hand, frozen feature extractor-based methods keep the feature extractor fixed after training the initial task and continually update the classifier to incorporate new tasks. Prototypical networks [13] utilized frozen feature extractor to generate class prototypes, enabling classification by calculating the distances to the prototype representations of each class. FeTrIL [15] generated pseudo-features for old classes by applying a geometric shift to new class features based on the difference between new and old class prototypes, which enabled the joint training of the classification head with both new and old class features. FeCAM [14] emphasized the greater heterogeneity in the distribution of new classes compared to old ones and classified the features based on the anisotropic Mahalanobis distance to the prototypes after the first task. To bridge the significant distribution gap between the real and the recorded features, DiffFR [18] constructed diffusion models to generate features that closely resembled the real features and reconstructed them to enhance the robustness of the classifier.

## 2.2 Self-Supervised Learning

In recent years, self-supervised learning (SSL) has proven to be an effective approach for learning general representations from unlabeled data. This is accomplished through predefined proxy tasks, such as rotation prediction [16], patch permutation [29], image colorization [30], inpainting [31], and clustering [32]. These tasks help the model acquire task-agnostic features, enabling it to generalize across various downstream tasks like few-shot learning [33] and semi-supervised learning [34], to improve the model robustness [35], class imbalance [36], etc. Recently, SLA [17] demonstrated that SSL can enhance supervised classification performance by augmenting the original labels through input transformations.

In NECIL, PASS [9] employed rotation self-supervision to learn generalizable and transferable features for future tasks. SASS [19] introduced an auxiliary classifier after each ResNet block to perform self-supervised rotation classification, which facilitated the learning of more generalized representations and achieved a better stability-plasticity trade-off. HRFSN [20] used random positive samples to perform rotation prediction tasks, enabling the feature extractor to learn richer feature representations through complex rotation prediction tasks. DiffFR [18] integrated SSL with instance similarity constraints to train the feature extractor, reducing feature overlaps by applying class- and instance-level discrimination constraints. Our *CDSSL* is implemented in a manner similar to the SSL in PASS [9]. Building upon this, we guide the model to learn task-agnostic classes that are highly ambiguous, thereby facilitating the extraction of more discriminative representations.

## 2.3 Growing Architectures and Ensemble Leaning

Architecture-based methods [37, 38] design task-specific parameters to adapt to new tasks while maintaining performance on previous ones. Although these approaches substantially enhance performance, the linear increase in parameters as the number of tasks increases becomes unsustainable. Moreover, the effective integration of knowledge across tasks remains a significant challenge. To address these issues, CoSCL [23] proposed an ensemble composed of a limited number of experts and penalized the discrepancies in their feature representation predictions to promote cooperation. SEED [39] developed a selective training ensemble of experts to mitigate forgetting, while leveraging the diverse expertise of the specialists for joint prediction. However, the newly created experts in this method lead to a significant decline in performance, causing the combined predictive capability of the expert ensemble to be inferior to that of a single model. Therefore, this paper explores solutions to address this issue.

## 3 Method

In this section, we first define the Non-Exemplar Class Incremental Learning (NECIL) scenario. Then, we outline the training and inference process of the proposed *CLOVER*. Finally, we provide a detailed introduction of the proposed confusion-driven self-supervised learning (*CDSSL*) method and the progressively weighted prediction (*PWP*) strategy.

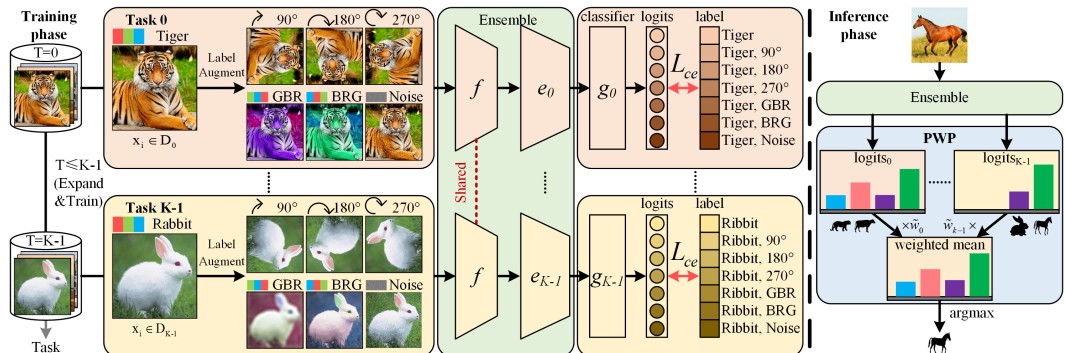

Figure 2: Overview of our proposed *CLOVER*. For tasks $t \leq K - 1$, the classes of the current task are augmented to generate task-agnostic classes using transformations such as rotation, color channel swapping, and noise injection. A new expert ($f \circ e_t$) is subsequently introduced and trained on this combined set of classes to enhance representation extraction capabilities and mitigate representation overlap between new and old classes. During the inference phase, the input image is processed through the ensemble model to generate expert-specific logits. These logits are then combined using Eq. (2) to calculate the weighted mean logits, with the weights for each expert determined by the proposed *PWP* strategy. Finally, the class prediction is obtained by performing the argmax operation on these logits.

## 3.1 Preliminaries

NECIL emphasizes the capability to sequentially acquire knowledge across a series of unrelated tasks without retaining or revisiting previously encountered examples. The training data for task $t \in \{0, 1, 2, \ldots, T\}$ is defined as $D_t = \{X_t, Y_t\} = \{x_t^i, y_t^i\}_{i=1}^{N_t}$, where $N_t$, $x_t^i$ and $y_t^i \in C_t$ represent the number of training images, a training image and the corresponding label for task $t$, respectively. Here, $C_t$ is the set of classes associated with task $t$, and the class sets for each task are disjoint, *i.e.*, $C_0 \cap C_1 \cap \cdots \cap C_T = \emptyset$. The model is evaluated on the combined test sets $Z_{0 \sim t} = Z_0 \cup Z_1 \cup \cdots \cup Z_t$, with the objective of accurately predicting both the classes of a new task and those of previously learned tasks, where $C_{0 \sim t} = C_0 \cup C_1 \cup \cdots \cup C_t$.

## 3.2 Overall Framework

As illustrated in Fig. 2, *CLOVER* is an ensemble model comprising $K$ experts, including a shared encoder $f$, $K$ expert encoders $e_i$ and their corresponding linear classifier $g_i$ ($i = 0, 1, \ldots, K - 1$). Notably, $f$ is frozen after training the first task. For task $t \leq K - 1$, *CDSSL* is employed to train the expert $t$ corresponding to task. After training, we freeze $e_t$ and discard $g_t$. Next, each existing expert $k \leq t$ computes the representation of class $c \in C_t$ and models it using a distinct Gaussian distribution $G(\mu_k^c, \Sigma_k^c)$, which is preserved and employed for classification in a manner similar to that of a prototype.

The inference process is similar to that of SEED [39]. The input $x$ is passed through each expert to obtain the expert-specific representation $r_k = e_k(f(x))$. We then compute the log-likelihood of $r_k$ with respect to the distribution of each class within that expert:

$$l_k^c(x) = -\frac{1}{2}[\ln(|\sum\nolimits_k^c|) + S \ln(2\pi) + (r_k - \mu_k^c)^T (\sum\nolimits_k^c)^{-1}(r_k - \mu_k^c)], \tag{1}$$

where $S$ is the representation dimension. It is worth noting that, since the new expert $h$ (where $h > 0$) does not have access to data from previous classes and cannot compute their representation distributions, we set $l_h^i(x) = -\infty$ for $i \in C_{0 \sim h-1}$ to indicate that predictions for these previous classes are impossible. Subsequently, we apply the softmax function to these values $\overline{l_k^1}, \overline{l_k^2}, \ldots, \overline{l_k^{|C|}} = softmax(l_k^1, l_k^2, \ldots, l_k^{|C|}; \tau)$ to obtain the predicted probabilities for each class within each expert, where $C$ is the set of all classes and $\tau$ represents the temperature. Finally, we calculate the weighted

sum of the predicted probabilities from all experts by:

$$\bar{l}(x) = \left\{ \sum_{k=0}^{M_i-1} \tilde{w}_k \times \bar{l}_k^i \right\}_{i=1}^{|C|}, \tag{2}$$

where $\tilde{w}_k = \frac{w_k}{\sum\limits_{j=0}^{M_i-1} w_j}$ and $M_i$ $(i = 1, 2, \ldots, |C|)$ represents the number of experts capable of predicting class $i$ (since each expert only retains the distributions of subsequent classes and performs predictions after completing its training). $w_k$ denotes the prediction weight calculated using the *PWP* method. Finally, class $c = argmax(\bar{l}(x))$ with the highest probability is selected as the final prediction.

### 3.3 Confusion-Driven Self-Supervised Learning

Considering a single expert, the feature extractor remains fixed following the initial training phase, and the distributions of the newly introduced classes are directly derived from the mapping of the frozen feature extractor. However, the absence of discriminative training between new and old classes increases the likelihood of confusion, causing the model to forget previous knowledge. To reduce the overlap of representations, we investigate the training methodology of feature extractor and propose the *CDSSL* strategy. This strategy focuses on training the model to differentiate between highly confusable task-agnostic classes, thereby enhancing its representation learning capabilities and reducing the overlap between representations of new and old classes.

Specifically, as illustrated in Fig. 2, we apply several transformations to the given image $x_i \in D_k$, including rotation, color channel swapping, and noise inception:

$$\tilde{x}_{7i+j} = \begin{cases} x_i, & j = 0 \\ ratate(x_i, j \times 90°), & j \leq 3 \\ GBR(x_i), & j = 4 \\ BRG(x_i), & j = 5 \\ x_i + s \times noise(0,1), & j = 6 \end{cases} \tag{3}$$

and assign new label $\tilde{y}_{7i+j} = 7y_i + j$, where $ratate(x_i,)$ refers to rotate $x_i$ by 90°, 180°, 270°. $GBR(\cdot)$ and $BRG(\cdot)$ indicate swapping $RGB$ channels to $GBR$ and $BRG$, respectively. $noise(0,1)$ represents a value randomly sampled from a standard normal distribution. $s$ denotes the noise intensity, which is typically set to 0.5. It generalizes the original N-way classification problem to a 7N-way classification problem. Next, we use cross-entropy loss to train the model:

$$L = L_{ce}(g_i(e_i(f(\tilde{x}))), \tilde{y}). \tag{4}$$

Comparing the widely used 4N-way self-supervised tasks, as demonstated in [9, 18], we generated new classes with higher levels of confusion by employing color channel swapping and noise inception. Thus, our proposed *CDSSL* encourages the model to learn more distinctive features, enabling it to more effectively differentiate between representations of new and old classes and thereby reducing their overlap. It is noteworthy that these task-agnostic classes are excluded during the inference phase and that their representation distributions across experts are neither computed nor stored.

### 3.4 Progressively Weighted Prediction

In NECIL, the base task typically constitutes half of the training data, with the remainder evenly distributed across incremental tasks. Consequently, due to the varying creation times of experts, the distributions of classes they cover differ significantly. Typically, newer experts encompass a narrower range of predictable classes. When such an expert encounters an old class outside its prediction scope, it often misclassifies it with high confidence. As shown in Fig. 1(c), such errors can mislead the equally weighted joint prediction system, ultimately resulting in inferior performance of the ensemble model compared to that of the individual models.

Therefore, considering that the number of predictable classes and the predictive ability vary across experts, we propose the progressively weighted prediction method. Specifically, we assign a low weight $\alpha$ to the new expert and progressively increase it by $\beta$, with the maximum weight being $\frac{1}{K}$.

Thus, the weight of each expert is defined as:

$$w_i = \begin{cases} 1 - \sum_{j=1}^{K-1} w_j, & i = 0 \\ \min\{\alpha + \beta \times (t - i), \frac{1}{K}\}, & i > 0 \end{cases} \tag{5}$$

where $t$ denotes the current task. As shown in Eq. (2), a weighted average of the predictions from each expert is then computed to derive the final outcome during joint prediction. In general, *PWP* accounts for the varying predictive capabilities of each expert by assigning distinct weights, thereby enhancing the joint prediction performance of the ensemble model in the early phase.

Table 1: Comparisons of the average accuracy and last accuracy (%) at different settings on CIFAR100, TinyImageNet, and ImageNet-Subset. CEAT [40] and FGKSR [41] are based on the ViT [42] structure, while all other methods are based on ResNet18 [43]. The result of SEED [39] is obtained using the author's official codebase, denoted by the asterisk (*). The best result is highlighted in **bold**, whereas the second-best method is indicated by underlining.

| Method | CIFAR100 | | | | | | TinyImageNet | | | | | | ImageNet-Subset | | | | | |
| | 5 tasks | | 10 tasks | | 20 tasks | | 5 tasks | | 10 tasks | | 20 tasks | | 5 tasks | | 10 tasks | | 20 tasks | |
| | Avg | Last | Avg | Last | Avg | Last | Avg | Last | Avg | Last | Avg | Last | Avg | Last | Avg | Last | Avg | Last |
|---|---|---|---|---|---|---|---|---|---|---|---|---|---|---|---|---|---|---|
| LwF_MC [44] | 45.9 | 36.1 | 27.4 | 17.0 | 20.1 | 15.9 | 29.1 | 17.1 | 23.1 | 12.3 | 17.4 | 8.8 | 34.9 | 24.1 | 31.2 | 20.0 | 27.5 | 17.4 |
| PASS [9] | 63.5 | 55.7 | 61.8 | 49.0 | 58.1 | 48.5 | 49.6 | 41.6 | 47.3 | 39.9 | 42.1 | 32.8 | 63.1 | 52.6 | 61.8 | 50.4 | 55.2 | 46.1 |
| SSRE [10] | 65.9 | 56.3 | 65.0 | 55.0 | 61.7 | 50.5 | 50.4 | 41.7 | 48.9 | 39.9 | 48.2 | 39.8 | 69.5 | 58.5 | 67.7 | 57.5 | 61.2 | 50.1 |
| FeTrIL [15] | 66.3 | - | 65.2 | 56.3 | 61.5 | - | 54.8 | - | 53.1 | - | 52.2 | - | 72.2 | - | 71.2 | - | 67.1 | - |
| PRAKA [12] | 70.0 | 61.6 | 68.9 | 60.4 | 65.9 | 56.2 | 53.3 | 46.4 | 52.6 | 45.2 | 49.8 | 40.6 | - | - | 69.0 | 61.3 | - | - |
| POLO [26] | 69.0 | - | 68.0 | - | 65.7 | - | 54.9 | 47.0 | 53.4 | 45.3 | 49.9 | 40.4 | 70.8 | 59.5 | 69.1 | 57.9 | - | - |
| TASS [45] | 68.8 | 59.3 | 67.4 | 57.9 | 62.8 | 53.8 | 55.1 | 44.1 | 54.2 | 43.9 | 52.8 | 43.6 | 74.3 | 63.1 | 72.6 | 57.9 | 68.8 | 57.6 |
| FGKSR [41] | 68.2 | 59.0 | 70.1 | 57.9 | 66.9 | 54.3 | 54.9 | 45.0 | 52.7 | 43.4 | 51.7 | 41.9 | - | - | 70.2 | 61.4 | - | - |
| CEAT [40] | 71.1 | - | 70.0 | - | 66.1 | - | 58.3 | 50.4 | 57.4 | 49.4 | 56.8 | 48.0 | 76.9 | 67.4 | 75.9 | 66.3 | 71.5 | 60.1 |
| SEED* [39] | 71.1 | 66.3 | 69.9 | 65.0 | 68.2 | 61.4 | 54.7 | 50.6 | 54.5 | 50.0 | 53.9 | 48.9 | 75.0 | 70.3 | 73.6 | 68.4 | 71.1 | 63.8 |
| FeCAM [14] | 70.9 | 62.1 | 70.8 | 62.1 | 69.4 | 58.5 | 59.6 | 52.8 | 59.4 | 52.8 | 59.3 | 52.8 | 78.3 | 70.9 | 78.2 | 70.9 | 75.1 | 66.3 |
| *CLOVER (Ours)* | **72.7** | **68.0** | **72.3** | **67.5** | **71.0** | **64.9** | **60.2** | **56.0** | **59.9** | **54.1** | 58.5 | 52.8 | 77.8 | **73.2** | 77.1 | **71.5** | 74.5 | **67.5** |

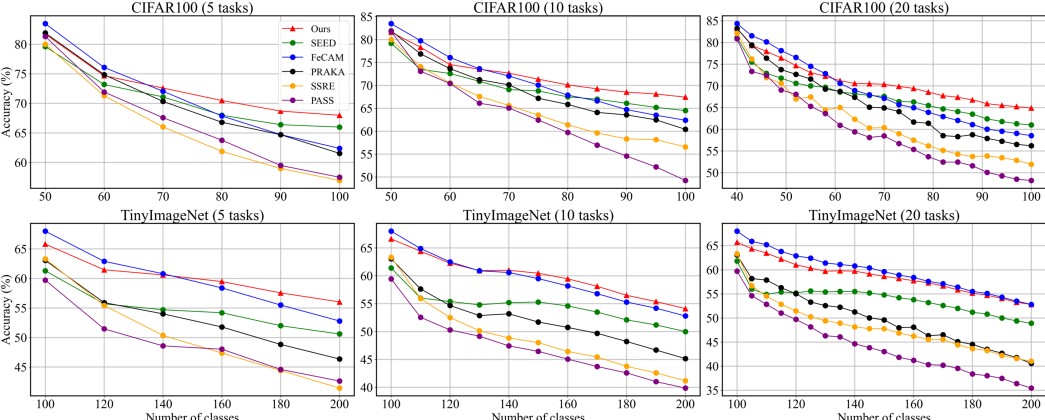

Figure 3: Illustration of the classification accuracy changes as tasks are being learned on CIFAR100 and TinyImageNet, which contains the complete curves. Precise data of our method is presented in Appendix B.

## 4 Experiments

### 4.1 Experimental Setup

**Datasets**. We conduct extensive experiments on three benchmark NECIL datasets: CIFAR100 [46], TinyImageNet [47], and ImageNet-Subset [48]. The CIFAR100 dataset comprises 100 classes, each with 600 color images at a resolution of $32 \times 32$ pixels, including 500 images for training and 100 images for testing. TinyImageNet is a subset of the ImageNet [48] designed for image classification, consisting of 200 classes, with each class containing 500 training images, 50 validation images, and 50 test images, all sized at $64 \times 64$ pixels. The ImageNet-Subset dataset, comprising 100 classes

Table 2: Ablation study of each component of *CLOVER* on CIFAR100 and TinyImageNet. The reported metric is average accuracy (%).

| Component | | | CIFAR100 | | | TinyImageNet | | |
|---|---|---|---|---|---|---|---|---|
| Baseline | *CDSSL* | *PWP* | 5 | 10 | 20 | 5 | 10 | 20 |
| ✓ | | | 69.9 | 69.7 | 67.7 | 56.3 | 55.3 | 54.5 |
| ✓ | ✓ | | 72.1 | 71.7 | 70.5 | 59.9 | 59.4 | 58.1 |
| ✓ | | ✓ | 70.5 | 70.2 | 68.2 | 56.5 | 55.8 | 54.9 |
| ✓ | ✓ | ✓ | **72.7** | **72.3** | **71.0** | **60.2** | **59.9** | **58.5** |

selected from ImageNet, contains about 13,000 training images and 50 test images per class, all standardized to a resolution of 224×224 pixels. We follow the configuration from PASS [9] and divide each dataset into three incremental settings: 5, 10, and 20 tasks.

**Evaluation Metrics**. We report two common metrics for CIL: the average accuracy and last accuracy. The former evaluates overall performance during incremental learning by calculating the average accuracy across all incremental phases. The latter assesses comprehensive performance upon completion of all tasks, emphasizing its validity in mitigating forgetting and integrating knowledge.

**Implementation Details and Reproducibility**. We use ResNet18 [43] as the backbone (same as SEED [39]). During the training phase, a linear classification head is employed, while a Bayesian classifier is used in the inference phase. The number of experts $K$ is set to 5. During training, the batch size is set to 128 and the model is optimized by the SGD optimizer with an initial learning rate 0.1 and weight decay 1e-4. The learning rate is multiplied by 0.1 at epochs 60, 120 and 160. All experiments are repeated three times and the average results are reported. The baseline is based on SEED [39], trained on the first 5 tasks, and shares the parameters of the first 5 layers across all expert models. Based on the aforementioned settings, all algorithms can be trained on a single NVIDIA A100 GPU.

## 4.2 Benchmark Comparison

Table 1 compares *CLOVER* with state-of-the-art NECIL methods on the CIFAR100, TinyImageNet, and ImageNet-Subset datasets. On CIFAR100, our *CLOVER* significantly outperforms the state-of-the-art methods across 5, 10, and 20 tasks, achieving higher average accuracy of 1.6%, 1.5%, and 1.6%, as well as improved last accuracy of 1.7%, 2.5%, and 3.5%, respectively. For 5 and 10 tasks on TinyImageNet, *CLOVER* exhibits superior average and last accuracy compared to the leading competing method. However, when the number of tasks increases to 20, FeCAM achieves better average accuracy. A similar trend is observed across all configurations on the ImageNet-Subset. Nonetheless, *CLOVER* achieves the highest last accuracy among all methods evaluated, underscoring its effectiveness in mitigating catastrophic forgetting while maintaining an optimal balance between plasticity and stability. Fig. 3 presents the accuracy variation curves on CIFAR100 and TinyImageNet. Although *CLOVER* shows comparatively lower performance in early tasks, it consistently outperforms other methods in last accuracy across all task settings on all three datasets. This demonstrates *CLOVER*'s superior ability to preserve knowledge during incremental training, establishing it as a highly competitive method even in long task settings.

## 4.3 Ablation Study

**Ablation Study of Different Components**. As presented in Table 2, we conduct ablation studies on CIFAR100 and TinyImageNet across the all settings (5, 10 and 20 tasks). The results indicate that *CDSSL* is a critical component, contributing to performance gains of 2.2%, 2.0% and 2.8% on CIFAR100, as well as 3.6%, 4.1%, 3.6% on the TinyImageNet. Furthermore, incorporating *PWP* yields an additional performance enhancement of approximately 0.5%. *PWP* complements *CDSSL* by improving the collaborative inference ability of the multi-expert model, thereby offsetting imbalances between experts and enhancing the model's overall complementarity. Collectively, our method achieves overall improvements range from 2–3% on CIFAR100 and 3–5% on TinyImageNet compared to the baseline. Moreover, *CDSSL* exhibits significant performance advantages in longer task sequences, highlighting the effectiveness of our approach in mitigating catastrophic forgetting.

**Ablation Study of *CDSSL***. To verify the effectiveness of each transformation in *CDSSL*, an ablation experiment is conducted on CIFAR100 and TinyImageNet in Table 3. The incorporation of rotation

Table 3: Ablation study of each transformation in the *CDSSL* on the CIFAR100 and TinyImageNet. Each transformation is applied independently without combinations or overlaps. The reported metric is average accuracy (%).

| Method | | | | CIFAR100 | | | TinyImageNet | | |
|---|---|---|---|---|---|---|---|---|---|
| Baseline | Rot | Color | Noise | 5 | 10 | 20 | 5 | 10 | 20 |
| ✓ | | | | 69.9 | 69.7 | 67.7 | 56.3 | 55.3 | 54.5 |
| ✓ | ✓ | | | 70.7 | 70.8 | 69.2 | 57.2 | 55.9 | 54.9 |
| ✓ | | ✓ | | 70.5 | 70.0 | 68.3 | 56.4 | 55.3 | 54.6 |
| ✓ | | | ✓ | 70.3 | 69.9 | 68.1 | 56.4 | 55.4 | 54.6 |
| ✓ | ✓ | ✓ | | 71.6 | 71.3 | 70.1 | 59.2 | 58.3 | 57.2 |
| ✓ | ✓ | | ✓ | 71.8 | 71.3 | 70.1 | 59.3 | 58.0 | 57.0 |
| ✓ | | ✓ | ✓ | 71.1 | 70.3 | 68.5 | 56.7 | 55.6 | 55.0 |
| ✓ | ✓ | ✓ | ✓ | **72.1** | **71.7** | **70.5** | **59.9** | **59.4** | **58.1** |

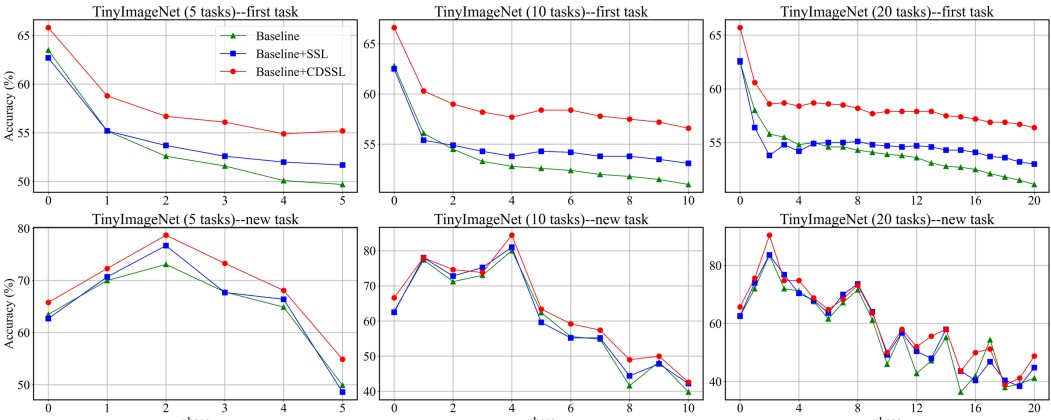

Figure 4: Accuracy of first/new task at each phase. The first row illustrates the variation in classification accuracy for the first task throughout the training process. The second row presents the classification accuracy of new tasks at various stages of training. Our *CDSSL* method enables more effective learning of new tasks while minimizing the impact on existing knowledge.

transformations yields average accuracy improvements of 0.8%, 1.1%, and 1.5% on CIFAR100, and 0.9%, 0.6%, and 0.4% on TinyImageNet, reflecting the performance gains achieved through rotation-based SSL methods commonly employed in existing studies. However, as illustrated in Fig. 1(a), the rotational transformation significantly alters the distribution of class representations, making it difficult for the model to learn highly discriminative features from such a simple task. To address this limitation, we propose an extension to the rotation-based SSL method by incorporating color channel swapping and noise inception transformations within our *CDSSL*. This strategy generates new classes that closely resemble their original counterparts, allowing the model to improve its representation learning by distinguishing between these highly similar classes. Specifically, the integration of color channel swapping resulted in additional average accuracy gains of 0.9%, 0.5%, and 0.9% on CIFAR100, alongside improvements of 2.0%, 2.4%, and 2.3% on TinyImageNet. Moreover, further incorporating noise inception transformations contributed an additional performance boost of at least 0.4%. Overall, we demonstrate both the effectiveness and necessity of each transformation employed in *CDSSL* while highlighting its substantial superiority over conventional SSL methods.

### 4.4 Representation Quality and Generalization Performance Comparison with SSL

In Fig. 4, we compare the effects of *CDSSL*, SSL, and baseline on the accuracy of the first task and new tasks during incremental training. The results indicate that *CDSSL* outperforms the other methods, significantly improving the accuracy of the first task while maintaining high performance for new tasks. In contrast, although SSL also achieves comparable results, its performance is markedly inferior to that of *CDSSL*. These findings suggest that *CDSSL* offers superior representational capabilities in incremental learning tasks compared to SSL, facilitating more effective category differentiation and is more suitable for NECIL.

### 4.5 Visualization

Fig. 5 shows a t-SNE visualization [49] of the dynamic changes in the representation space of the first expert during the early stages of training on CIFAR100. Specifically, we visualize the feature representations of visible classes at two distinct time points: 1) after training on 50 classes and adding 2 new classes, and 2) after training on 50 classes and adding 4 new classes. The introduction of 2 new classes leads to a slight overlap between the representation distributions of new and old classes within the Baseline. As more new classes are introduced, SSL struggles to effectively reduce the overlap between new and old classes, whereas *CDSSL* maintains a clear separation. Furthermore, *CDSSL* enhances the cohesion within each class's distribution, indicating that the model has acquired more representative and discriminative features. This improvement reflects an enhanced capability for feature extraction, which is essential for addressing inter-class overlap.

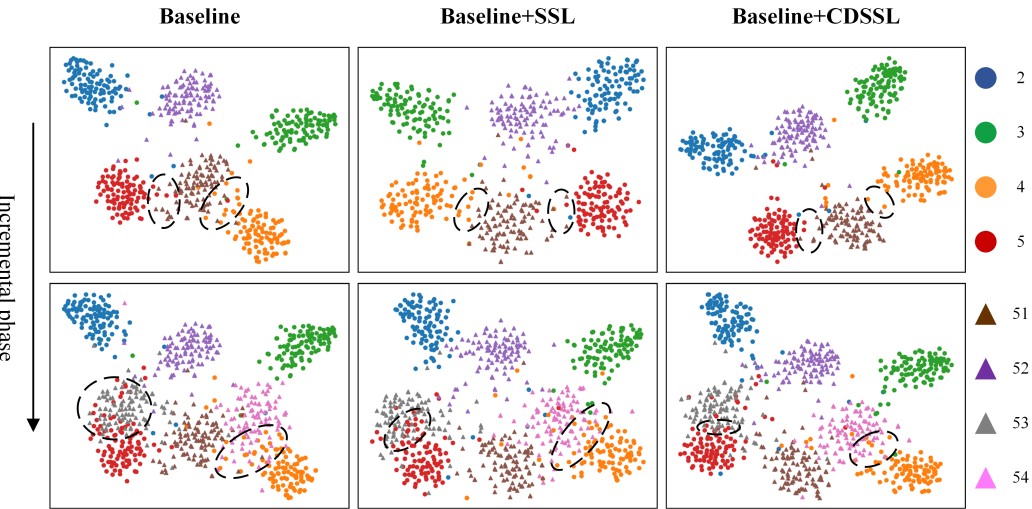

Figure 5: The visualization illustrates the distribution of class representations following the application of SSL and *CDSSL*, respectively. Initially, the model was trained on the first task containing 50 classes (denoted by circles). Subsequently, new classes (denoted by triangles) were incrementally introduced at each step, with the progression depicted from top to bottom.

## 5 Conclusion

In this paper, we develop *CLOVER*, an innovative framework for NECIL that aims to minimize the overlap between new and old classes within the representation space. Within this framework, we propose a *CDSSL* approach that not only employs rotation transformations commonly utilized in traditional self-supervised learning methods but also generates highly confusable task-agnostic classes through color channel swapping and noise injection. The model is subsequently trained to differentiate these classes, significantly enhancing its feature extraction capabilities and thereby mitigating the overlap between representations of new and old classes. Furthermore, we introduce an ensemble model to improve the discrimination of unknown classes and then present a progressively weighted prediction strategy to address interference caused by newly introduced experts within the ensemble model. The quantitative and qualitative results on three widely used datasets have demonstrated that our method achieves state-of-the-art performance, particularly excelling in improving last accuracy. It is worth noting that both plasticity and stability are crucial in NECIL, and thus, developing a method that can enhance both simultaneously is of substantial value. The proposed *CDSSL* introduces confusing classes to increase the training difficulty, thereby improving the model's discriminative capability between representations of new and old classes. This enables the model to maintain strong performance across both unknown and previously learned classes. Such an approach by enhancing training difficulty through the introduction of confusing classes offers an effective and promising direction for advancing non-exemplar class-incremental learning, and it holds potential for further extension to more complex or large-scale incremental learning scenarios in future research.

## Acknowledgments

This work was supported in part by the National Natural Science Foundation of China under Grants 62372170, 62272404, and 62502419, in part by the Natural Science Foundation of Hunan Province of China under Grant 2023JJ40638, and in part by the Research Foundation of Education Department of Hunan Province of China under Grant 23A0146.

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

# Appendix

In the appendix of this paper, Section A provides a detailed introduction to the evaluation metrics. Subsequently, Section B offers the detailed values of the accuracy curves on CIFAR100 and TinyImageNet datasets. Section C presents a more comprehensive and detailed ablation study. Section D comprehensively demonstrates the advantages of *CDSSL* over SSL from multiple perspectives, such as performance and visualization. Section E conducts an analysis of the model's scalability and investigates the rationale behind *CDSSL*'s adoption of three specific transformations: rotation, color channel permutation, and noise injection. Finally, Section F discusses the potential limitations and the broad impacts of our method.

## A  Evaluation Metrics

We utilize two evaluation metrics commonly applied in NECIL to assess the performance of different models under various settings. Each metric is defined formally below.

**Average Accuracy.** $Avg$ is defined as the average accuracy across all incremental stages, providing a comprehensive metric for fairly evaluating the overall incremental performance of different methods. Letting $a_{n,m}$ represents the accuracy of task $n$ after the training of phase $m$. The average accuracy is expressed as follows:

$$Avg = \frac{1}{T+1} \sum_{m=0}^{T} \sum_{n=0}^{m} a_{n,m}. \tag{6}$$

**Last Accuracy.** $Last$ evaluates the model's final performance on the validation sets of all previously encountered tasks, serving as a key metric for assessing its ability to mitigate catastrophic forgetting, integrate knowledge across tasks, and maintain robust generalization. Last accuracy is defined as follows:

$$Last = \sum_{i=0}^{T} a_{i,T}. \tag{7}$$

## B  Detailed Values of the Accuracy Curves

To facilitate comparisons in future research, we present the detailed values of the accuracy curves (as depicted in Fig. 3 of the main text) in Tables 4, 5 and 6.

Table 4: Detailed values of classification accuracy under the setting of 5 tasks.

| Dataset | Phase | | | | | |
|---|---|---|---|---|---|---|
| | 0 | 1 | 2 | 3 | 4 | 5 |
| CIFAR100 | 81.70 | 74.68 | 72.6 | 70.49 | 68.70 | 67.99 |
| TinyImageNet | 65.82 | 61.47 | 60.60 | 59.48 | 57.56 | 56.02 |

Table 5: Detailed values of classification accuracy under the setting of 10 tasks.

| Dataset | Phase | | | | | | | | | | |
|---|---|---|---|---|---|---|---|---|---|---|---|
| | 0 | 1 | 2 | 3 | 4 | 5 | 6 | 7 | 8 | 9 | 10 |
| CIFAR100 | 81.62 | 78.33 | 74.55 | 73.53 | 72.69 | 71.36 | 70.13 | 69.26 | 68.65 | 68.12 | 67.45 |
| TinyImageNet | 66.62 | 66.40 | 62.27 | 60.94 | 61.00 | 60.47 | 59.46 | 58.13 | 56.51 | 55.39 | 54.12 |

Table 6: Detailed values of classification accuracy under the setting of 20 tasks.

| Dataset | Phase | | | | | | | | | |
|---|---|---|---|---|---|---|---|---|---|---|
| | 0 | 1 | 2 | 3 | 4 | 5 | 6 | 7 | 8 | 9 |
| CIFAR100 | 83.25 | 79.33 | 77.96 | 77.47 | 74.69 | 73.07 | 72.24 | 71.21 | 70.56 | 70.51 |
| TinyImageNet | 65.72 | 64.36 | 63.47 | 62.24 | 61.03 | 60.35 | 59.70 | 59.78 | 59.73 | 59.16 |

| Dataset | Phase | | | | | | | | | |
|---|---|---|---|---|---|---|---|---|---|---|
| | 10 | 11 | 12 | 13 | 14 | 15 | 16 | 17 | 18 | 19 | 20 |
| CIFAR100 | 70.37 | 69.89 | 69.42 | 68.57 | 67.76 | 67.39 | 66.78 | 65.91 | 65.51 | 65.23 | 64.90 |
| TinyImageNet | 58.64 | 58.21 | 57.74 | 57.30 | 56.65 | 55.84 | 55.17 | 54.71 | 54.05 | 53.25 | 52.80 |

## C  More Ablation Study

### C.1  Components Ablation for ImageNet-Subset

To further verify the generalizability of *CLOVER*, we conduct components ablation on the ImageNet-Subset dataset, as illustrated in Table 7.

Table 7: Ablation study of each component of *CLOVER* on ImageNet-Subset. The reported metric is average accuracy (%).

| Component | | | ImageNet-Subset | | |
|---|---|---|---|---|---|
| Baseline | *CDSSL* | *PWP* | 5 | 10 | 20 |
| ✓ | | | 75.1 | 73.9 | 71.1 |
| ✓ | ✓ | | 77.7 | 76.8 | 74.3 |
| ✓ | ✓ | ✓ | **77.8** | **77.1** | **74.5** |

### C.2  Ablation Study of *CDSSL* for ImageNet-Subset

We conduct ablation studies on the various transformations of *CDSSL* on the ImageNet-Subset dataset. As presented in Table 8, the diverse transformations within *CDSSL* yield substantial performance improvements, confirming the effectiveness of our *CDSSL* approach on large-scale datasets.

Table 8: Ablation study of each transformation in the *CDSSL* on ImageNet-Subset. The reported metric is average accuracy (%).

| Method | | | | ImageNet-Subset | | |
|---|---|---|---|---|---|---|
| Baseline | Rot | Color | Noise | 5 | 10 | 20 |
| ✓ | | | | 75.1 | 73.9 | 71.1 |
| ✓ | ✓ | | | 76.7 | 75.6 | 73.4 |
| ✓ | ✓ | ✓ | | 77.4 | 76.5 | 73.9 |
| ✓ | ✓ | ✓ | ✓ | **77.7** | **76.8** | **74.3** |

### C.3  Number of Shared Layers

In Table 9, we compare the performance of *CLOVER* under different parameter configurations (adjusted by varying the number of shared layers $f$) with SEED [39], which is likewise an ensemble-based model. The first expert, extensively trained on the initial task with abundant data, demonstrates strong representation extraction capabilities. By sharing certain layers with the first expert, the representation extraction capabilities of the other experts are enhanced, leading to an overall improvement in *CLOVER*'s performance. Notably, the best results are achieved when 5 layers are shared. Furthermore, *CLOVER* maintains robust performance even with reduced parameter counts, consistently outperforming SEED. These results strongly validate the effectiveness of the proposed method.

Table 9: Comparisons of the average accuracy and last accuracy (%) with various shared layers on CIFAR100.

| Method | #Params | CIFAR100 | | | | | |
| | | 5 tasks | | 10 tasks | | 20 tasks | |
| | | Avg | Last | Avg | Last | Avg | Last |
|---|---|---|---|---|---|---|---|
| SEED* [39] | 56.0M | 71.1 | 66.3 | 69.9 | 65.0 | 68.2 | 61.4 |
| *CLOVER*(0 shared) | 56.0M | 72.6 | 67.5 | 72.1 | 66.8 | 71.0 | 64.5 |
| *CLOVER*(5 shared) | 55.4M | **72.7** | **68.0** | **72.3** | **67.5** | **71.0** | **64.9** |
| *CLOVER*(9 shared) | 53.3M | 72.6 | 67.9 | 72.2 | 67.3 | 70.9 | 64.6 |
| *CLOVER*(13 shared) | **44.9M** | 72.4 | 67.5 | 72 | 66.9 | 70.4 | 64.1 |

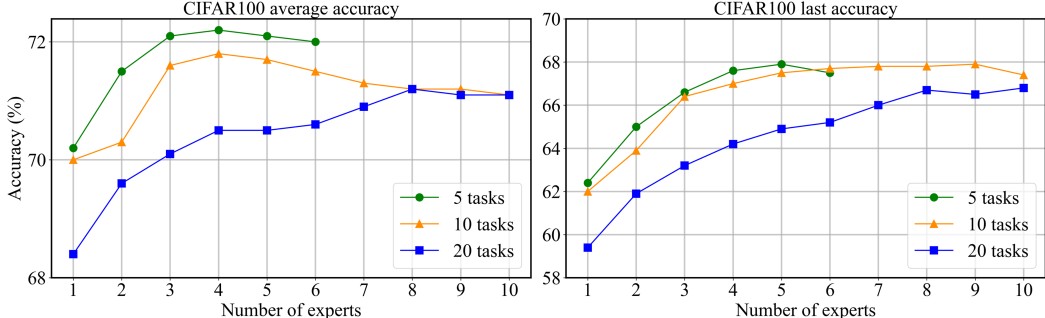

Figure 6: The impact of the number of experts in *CLOVER* on both the average accuracy and last accuracy.

## C.4   Number of Experts

In Fig. 6, we investigate the impact of the number of experts on the average accuracy and the last accuracy attained by *CLOVER*. As the number of experts increases from 1 to 4, *CLOVER* exhibits consistent improvements in both average accuracy and last accuracy on 5, 10, and 20 task settings. However, as the number of experts continues to increase, the average accuracy of *CLOVER* in 5 and 10 tasks scenarios exhibits a declining trend, with no significant enhancement observed in the last accuracy. While additional performance improvements can be achieved in 20 tasks setting, these enhancements are not uniformly effective across all configurations and result in a substantial rise in parameter count. Consequently, we ultimately adopt *CLOVER* with five experts (*i.e.*, $K$=5), following the methodology outlined by SEED [39], to ensure a fair and equitable comparison.

## C.5   The Sensitivity of Hyper-parameter $\alpha$ and $\beta$

In *PWP*, the parameters $\alpha$ and $\beta$ play a critical role in determining weight allocation. To investigate their impact, we conduct a sensitivity analysis on CIFAR100 and TinyImageNet, as presented in Table 10. Given that the maximum weight for a new expert is constrained to 0.2 (as $K = 5$), we vary $\alpha$ within the range 0.15, 0.10, 0.05 and allow the weight to reach its maximum either after one task or two tasks. Notably, a higher initial $\alpha$ yields better performance in short-task settings, while a lower initial $\alpha$ is advantageous in long-task scenarios. This distinction arises from the fact that, in long-task settings, newly introduced experts have access to less training data, leading to less reliable predictions that can be effectively mitigated by assigning them lower weights. Overall, *PWP* demonstrates robustness to variations in $\alpha$ and $\beta$, consistently achieving significant improvements.

## D   Further Comparison with SSL

### D.1   Representation Quality and Generalization Performance Comparison

To assess the effectiveness of *CDSSL*, we conduct a comparative analysis of its impact on the accuracy of both the first task and the new tasks, in comparison with SSL and the baseline, across the CIFAR100 and ImageNet-Subset datasets during incremental training, as illustrated in Fig. 7. The proposed

*CDSSL* achieves higher accuracy on both the first task and new task across all evaluated datasets, highlighting the generalizability of its improved representation extraction capabilities.

Table 10: Robustness testing of hyper-parameters $\alpha$ and $\beta$ on CIFAR100 and TinyImageNet. The first row, where $\alpha$ and $\beta$ equal to $-$, represents the results obtained without *PWP*. The reported metric is average accuracy (%).

| Parameters | | CIFAR100 | | | TinyImageNet | | |
|---|---|---|---|---|---|---|---|
| $\alpha$ | $\beta$ | 5 | 10 | 20 | 5 | 10 | 20 |
| $-$ | $-$ | 72.1 | 71.7 | 70.5 | 59.9 | 59.4 | 58.1 |
| 0.15 | 0.05 | 72.7 | 72.0 | 70.9 | 60.2 | 59.9 | 58.5 |
| | 0.025 | 72.7 | 72.1 | 70.9 | 60.1 | 59.9 | 58.5 |
| 0.10 | 0.10 | 72.7 | 72.1 | 70.9 | 60.1 | 59.9 | 58.5 |
| | 0.05 | 72.7 | 72.3 | 71.0 | 59.9 | 59.9 | 58.6 |
| 0.05 | 0.15 | 72.7 | 72.2 | 71.0 | 59.8 | 59.9 | 58.6 |
| | 0.075 | 72.6 | 72.2 | 71.1 | 59.7 | 59.9 | 58.6 |

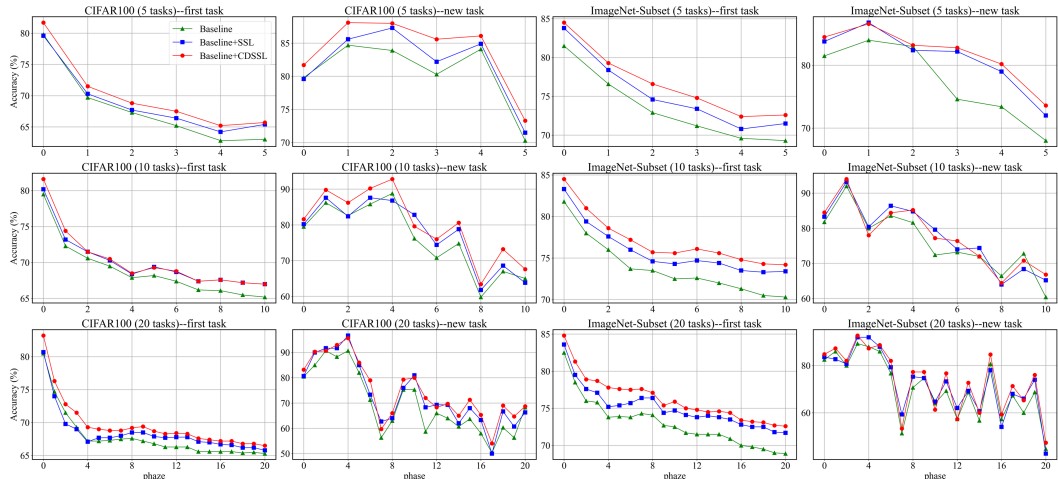

Figure 7: Accuracy of first/new task at each phase on CIFAR100 and ImageNet-Subset.

## D.2 Quantitative Comparison

To more clearly demonstrate the advantages of *CDSSL* over traditional SSL, Table 11 compares the performance of the Baseline enhanced with *CDSSL* versus that enhanced with traditional SSL. The experimental results show that *CDSSL* achieves superior performance across all settings. This improvement arises from the introduction of highly confusable task-agnostic classes in *CDSSL*, which enable the model to extract the most discriminative features for various class. Consequently, *CDSSL* significantly minimizes the overlap in representation between new and old classes, thereby improving the model's capacity for incremental learning.

## D.3 The Similarity Between Various Task-Agnostic Classes and the Raw Classes

In Fig. 8, we compare the KL divergence [50] between the raw classes and task-agnostic classes, generated through rotation, color channel swapping, and noise injection, on CIFAR100. A smaller

Table 11: Quantitative comparison between *CDSSL* and SSL. The reported metric is average accuracy (%).

| Method | CIFAR100 | | | TinyImageNet | | | ImageNet-Subset | | |
|---|---|---|---|---|---|---|---|---|---|
| | 5 | 10 | 20 | 5 | 10 | 20 | 5 | 10 | 20 |
| Baseline | 69.9 | 69.7 | 67.7 | 56.3 | 55.3 | 54.5 | 75.1 | 73.9 | 71.1 |
| Baseline+SSL | 70.7 | 70.8 | 69.2 | 57.2 | 55.9 | 54.9 | 76.7 | 75.6 | 73.4 |
| **Baseline+*CDSSL*** | **72.1** | **71.7** | **70.5** | **59.9** | **59.4** | **58.1** | **77.7** | **76.8** | **74.3** |

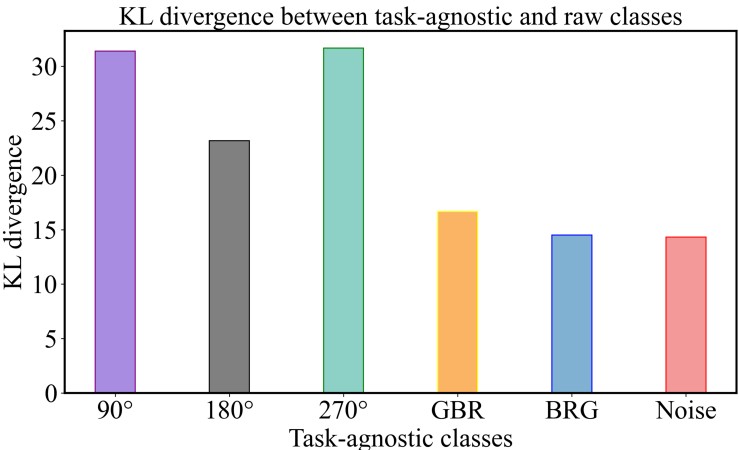

Figure 8: KL divergence between task-agnostic and raw classes on CIFAR100.

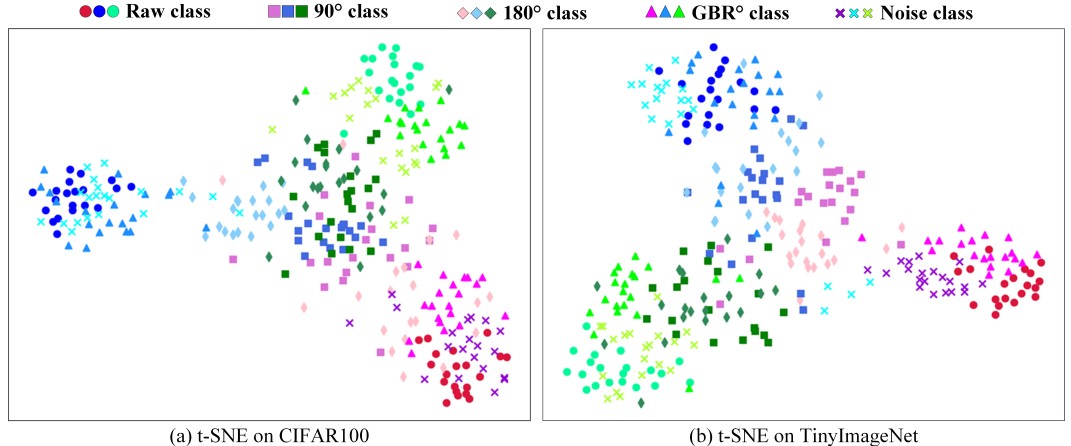

(a) t-SNE on CIFAR100         (b) t-SNE on TinyImageNet

Figure 9: t-SNE visualization of raw classes and task-agnostic classes on CIFAR-100 and TinyImageNet. Samples with similar colors correspond to the same raw class, while different marker shapes represent task-agnostic classes derived by applying distinct transformations.

KL divergence indicates greater similarity between the representation distributions, leading to a higher likelihood of confusion. Furthermore, Fig. 9 provides an intuitive visualization of the spatial relationships between the raw and task-agnostic classes. These results indicate that task-agnostic classes generated via color channel swapping and noise injection are more susceptible to confusion with the raw classes.

# E More Analysis

## E.1 Parameter Scalability Analysis

To better demonstrate scalability of *CLOVER*, Fig. 10 illustrates the variation in model parameters across incremental phases. Simply, throughout the training process from Task 0 to Task 4, the number of parameters increases by approximately 11.2M with each new task that arrives. Beyond this stage, no additional experts are incorporated, and the number of parameters remains stable at 55.4M.

## E.2 More Transformation Experiments

To validate the effectiveness and generalizability of the transformations employed in *CDSSL*, Table 12 presents a comparison of the performance improvements achieved by different transformations. The

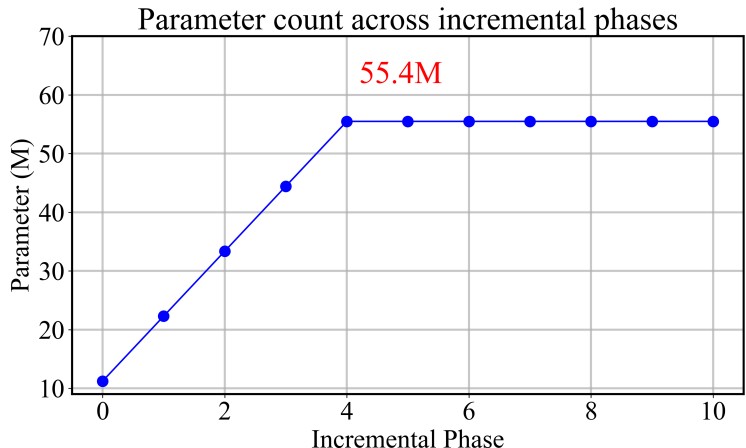

Figure 10: Variation of model parameter count across incremental phases.

Table 12: Comparison of model performance on CIFAR100 and TinyImageNet datasets after applying various transformations. "Blurring" applys 5×5 Gaussian blur kernels with standard deviations of 1.0. "CenterCrop" crops a 224×224 image to 192×192 and then resizes it back to 224×224. The reported metric is average accuracy (%).

| Method | CIFAR100 | | | TinyImageNet | | |
|---|---|---|---|---|---|---|
| | 5 | 10 | 20 | 5 | 10 | 20 |
| Baseline | 69.9 | 69.7 | 67.7 | 56.3 | 55.3 | 54.5 |
| Baseline+Rot | 70.7 | 70.8 | 69.2 | 57.2 | 55.9 | 54.9 |
| Baseline+Color | 70.5 | 70.0 | 68.3 | 56.4 | 55.3 | 54.6 |
| Baseline+Noise | 70.3 | 69.9 | 68.1 | 56.4 | 55.4 | 54.6 |
| Baseline+Blurring | 70.3 | 69.9 | 68.2 | 56.1 | 55.9 | 54.3 |
| Baseline+CenterCrop | 71.4 | 70.5 | 69.0 | 55.8 | 54.7 | 54.3 |

results indicate that not all transformations yield effective results, as some may perform well on CIFAR-100 while exhibiting poor performance on TinyImageNet. In contrast, the transformations in *CDSSL* (rotation, color channel swapping, and noise injection) consistently improve performance across datasets, demonstrating superior generalizability.

In addition, an excessive number of task-agnostic classes may lead to an overemphasis on task-agnostic knowledge, which can impede class recognition. To investigate this phenomenon, Table 13 demonstrates the results of *CDSSL* enhanced with blurring, cropping and more color channel swapping (*i.e.* swapping RGB channels to RBG, GRB and BGR) transformations and compares them with those produced by *CDSSL* alone. While these transformations improved performance on CIFAR-100, they adversely affected TinyImageNet due to its greater number of categories and finer details requiring more effective feature extraction.

Table 13: Comparison of model performance on the CIFAR-100 and TinyImageNet datasets after applying additional transformations beyond Baseline + *CDSSL*. "MoreColor" refers to the creation of new classes by swapping the RGB channels into RBG, GRB, and BGR. The reported metric is average accuracy (%).

| Method | CIFAR100 | | | TinyImageNet | | |
|---|---|---|---|---|---|---|
| | 5 | 10 | 20 | 5 | 10 | 20 |
| Baseline+*CDSSL* | 72.1 | 71.7 | 70.5 | 59.9 | **59.4** | **58.1** |
| Baseline+*CDSSL*+Blurring | 72.3 | 71.7 | **70.9** | 59.8 | 58.7 | 57.6 |
| Baseline+*CDSSL*+CenterCrop | **73.1** | **72.6** | **70.9** | **60.1** | 58.6 | 57.7 |
| Baseline+*CDSSL*+MoreColor | 72.4 | 71.9 | **70.9** | 59.7 | 58.5 | 58.0 |

Table 14: Comparison of the average accuracy (%) of different methods under the class-balanced setting.

| Method | CIFAR100 | | |
|---|---|---|---|
| | 5 | 10 | 20 |
| EFC [51] | - | 60.87 | 55.78 |
| SEED | 70.9 | 69.3 | 62.9 |
| Baseline | 71.1 | 66.7 | 60.3 |
| **Baseline+*CDSSL*** | **73.2** | **71.0** | **66.7** |

Table 15: Evolution of average Bhattacharyya distance during training under the 5-task setting on CIFAR-100.

| Method | | | Task | | | | | |
|---|---|---|---|---|---|---|---|---|
| Baseline | SSL | *CDSSL* | 0 | 1 | 2 | 3 | 4 | 5 |
| ✓ | | | 7.62 | 6.75 | 6.19 | 5.76 | 5.38 | 5.05 |
| ✓ | ✓ | | 10.17 | 9.34 | 8.85 | 8.41 | 8.06 | 7.74 |
| ✓ | | ✓ | 12.12 | 11.02 | 10.34 | 9.77 | 9.31 | 8.89 |

### E.3  Performance under the class-balanced setting

To further evaluate the effectiveness of *CDSSL* across different incremental learning configurations, we conduct experiments under the class-balanced setting, where each task contains an equal number of classes. As presented in Table 14, *CLOVER* exhibits strong performance and generalizability, with *CDSSL* achieving a 6.4% improvement in average accuracy under the long-task setting and consistently surpassing existing methods across all settings. These results further attest to the robustness and adaptability of *CDSSL* in diverse incremental learning scenarios.

### E.4  Quantitative Analysis of Inter-Class Distribution Confusion

To intuitively demonstrate the effectiveness of our method in enhancing inter-class separability, the Bhattacharyya distance [52, 53] is employed as a metric to quantify the overlap between two probability distributions, where a greater overlap corresponds to a smaller distance, while a lesser overlap yields a larger one. This metric effectively captures the degree of separability and the clarity of class boundaries. The Bhattacharyya distance is calculated as follows:

$$D_B = \frac{1}{8}(\mu_1 - \mu_2)^T \Sigma^{-1}(\mu_1 - \mu_2) + \frac{1}{2}ln(\frac{|\Sigma|}{\sqrt{|\Sigma_1||\Sigma_2|}}) \tag{8}$$

where $\mu_1, \mu_2$ denote the mean vectors of the two class distributions, $\Sigma_1$, $\Sigma_2$ are the corresponding covariance matrices, $\Sigma = \frac{1}{2}(\Sigma_1 + \Sigma_2)$ represents the average covariance matrix, and $|\cdot|$ denotes the determinant.

Based on this metric, we track the average class-boundary clarity throughout the training process under the 5-task setting of CIFAR-100, as summarized in Table 15. The results show that *CDSSL* effectively improves class separability by increasing the Bhattacharyya distance, thereby reducing confusion and overlap between class representation distributions.

## F  Limitations and Broader Impacts

**Limitations.** In this study, we propose a method for non-exemplar class incremental learning. However, the approach relies on a base task with abundant training data and has not yet been validated in a setting where all tasks contain an equal amount of data. Moreover, the training cost of our method is relatively high. Specifically, we extend the original N-way classification task to a 7N-way classification task, thereby increasing the training time. In addition, compared to single-model approaches, our method employs an ensemble model, which results in a larger number of parameters. To address the high training cost, future work will focus on developing strategies that reduce both training time and parameter size, aiming to achieve a more favorable balance between performance and computational efficiency.

**Broader Impacts.** Non-exemplar class incremental learning, which enables the continuous accumulation of knowledge without retaining old samples, is of considerable significance in the current era of growing data privacy concerns. Our proposed *CLOVER* framework fully leverages the advantage that frozen models are inherently resistant to catastrophic forgetting during updates, focusing on mitigating representation overlap between old and new classes. *CLOVER* synergistically combines self-supervised learning and ensemble learning to effectively and non-conflictually address this challenge, offering a novel pathway for future researchers to combat forgetting. Moreover, the idea of enhancing the representation extraction capabilities of frozen models to further alleviate representation overlap, as explored in this work, presents a promising direction for continued investigation and may foster further advancements in the field. Overall, our study advances the development of machine learning and introduces an innovative solution to the domain of non-exemplar class incremental learning.

