# OpenReview forum: "Confusion-Driven Self-Supervised Progressively Weighted Ensemble Learning for Non-Exemplar Class Incremental Learning"
_NeurIPS.cc/2025/Conference — NeurIPS 2025 poster_

### Official Review · Reviewer_KrMA · 2025-06-23

**Clarity:** 3
**Significance:** 2
**Originality:** 2
**Rating:** 3
**Confidence:** 4

**Summary:**

This paper presents CLOVER, a framework for Non-Exemplar Class Incremental Learning (NECIL) that addresses the representation overlap problem. It introduces two main components: Confusion-Driven Self-Supervised Learning (CDSSL), which enhances representation robustness via challenging augmentations, and Progressively Weighted Prediction (PWP), which gradually adjusts the influence of expert models. A shared encoder is frozen after the first task, and each new class is modeled with Gaussian distributions by task-specific experts. During inference, predictions are made via a Bayesian ensemble. Experiments on CIFAR100, Tiny ImageNet, and ImageNet-Subset show state-of-the-art performance.

**Questions:**

1.What’s the difference between your proposed method and SEED?

2.The training and inference process are demonstrated in Sec. 3.2, but the training process when t > K-1 wasn’t specified in the paper. Is it the same as SEED? Or stop training after learning the K-1 task? This should be clearly specified.

3.There’s only one loss function employed during the training stage. Are there any other constraints or regularizations that are utilized to mitigate catastrophic forgetting?

4.As shown in Table 1, CEAT and FGKSR are based on the ViT structure, while the other methods are based on ResNet18. Is this comparison fair? What is the comparison of the trainable parameters between CEAT, FGKSR and other ResNet18-based methods?

5.The evaluation metric that was employed in the experiments should be clearly specified in the paper. Which accuracy metric was used in Table 2 and Figure 4?

6.What about the performance of the proposed method on an equally split task setting?

**Ethical Concerns:**

["NO or VERY MINOR ethics concerns only"]

**Final Justification:**

Thank you for the detailed response from the authors. The feedback addresses some of my concerns. However, I remain unconvinced by several aspects. For instance, the performance drop with PSP in the equally split task setting is not well justified—the explanation that PSP is designed with a base task in mind is insufficient. Additionally, the novelty of the proposed approach appears limited, as similar ideas have been explored in previous work. Moreover, the training cost remains a significant concern, both in terms of time and GPU resources. Overall, I believe the paper still requires substantial improvement.

**Limitations:**

Yes

**Quality:**

2

**Strengths And Weaknesses:**

Strength:

1.This paper is generally well-written and well-motivated. It’s important and interesting to mitigate catastrophic forgetting from “representation overlap” perspective.

2.This paper proposed a method in NECIL which combined with SSL and ensemble learning, achieving state-of-the-art performance in the proposed evaluation protocol.

3.This paper offers extensive experimental analysis and results, demonstrating the effectiveness of the method and the significance of different components of the proposed method.

Weakness:

1.The method's reliance on a substantial amount of classes for the base task is a notable constraint, as its efficacy has not yet been validated in scenarios where all incremental tasks contain an equal amount of classes.

2.The novelty of this paper is limited: self-supervised learning, ensemble learning, and frozen feature extractors are established techniques in CIL.

3.This paper doesn’t provide a clear demonstration and detailed explanation of the learning process, especially the training stage.

4.The training cost of the proposed method is relatively high, stemming from the expansion of the original N-way classification problem to a 7N-way problem.

5.The comprehensiveness of the experimental comparisons could be further strengthened. Specifically, it would be beneficial to include empirical evaluations against several state-of-the-art NECIL approaches [a,b] which are published in recent years if possible.

[a] Magistri, Simone, Tomaso Trinci, Albin Soutif, Joost van de Weijer, and Andrew D. Bagdanov. "Elastic Feature Consolidation For Cold Start Exemplar-Free Incremental Learning." In The Twelfth International Conference on Learning Representations.

[b] Rypeść, Grzegorz, Sebastian Cygert, Tomasz Trzciński, and Bartłomiej Twardowski. "Task-recency bias strikes back: Adapting covariances in exemplar-free class incremental learning." Advances in Neural Information Processing Systems 37 (2024).

---

> ### Author Rebuttal · Authors · 2025-07-30
>
> We greatly appreciate your constructive and insightful comments. We address all weaknesses and questions below.
>
> **W1 & Q6: Performance validation on an equally split task setting.**
>
> In NECIL, two primary data partitioning strategies are employed. The first strategy, which is utilized in our work as well as by many others, involves a base task that encompasses a substantial number of classes, followed by incremental tasks that each contain an equal number of classes. The second strategy distributes the total number of classes evenly across all tasks without designating a specific base task. These two setups emphasize different aspects of the model’s performance. The former prioritizes the stability of previously acquired knowledge during incremental updates, while the latter emphasizes the model’s plasticity in acquiring new knowledge. These differences reflect distinct evaluation priorities in incremental learning.
>
> Table 1 reports the performance of our method under the class-balanced setting, demonstrating the effectiveness and generalizability of the proposed CDSSL. Notably, CDSSL achieves a 6.4% improvement in average accuracy under the long-task setting. In this configuration, each expert is trained on a same number of classes, resulting in relatively balanced predictive capabilities across experts. However, our proposed PWP strategy is specifically designed for scenarios involving a base task, which results in suboptimal performance within class-balanced settings. Nonetheless, it is important to emphasize that our method, which relies exclusively on CDSSL, surpasses several robust baselines. This further validate the robustness and adaptability of CDSSL across various incremental learning scenarios.
>
> **Table 1. Comparison of the average accuracy (%) of different methods under the class-balanced setting.**
>
> |Method|CIFAR100|||
> |:---:|:---:|:---:|:---:|
> ||5 Tasks|10 Tasks|20 Tasks|
> |EFC|-|60.87|55.78|
> |SEED|70.9|69.3|62.9|
> |Baseline|71.1|66.7|60.3|
> |Baseline+CDSSL|73.2|71|66.7|
> |Baseline+CDSSL+PWP|71.8|70.2|66.4 |
>
> **W2: The novelty of this paper is limited: self-supervised learning, ensemble learning, and frozen feature extractors are established techniques in CIL.**
>
> Although there are some works (e.g, PASS, SEED), we believe there are fundamental differences between our method and existing approaches.
>
> **First**, in traditional SSL methods such as PASS, the task-agnostic classes generated through rotation exhibit significant semantic differences from the raw classes (see Fig. 1a), which limits their effectiveness. In contrast, our CDSSL generates highly confusable task-agnostic classes, encouraging the model to distinguish them from the raw classes. This approach facilitates more effective discriminative feature extraction and reduces representation overlap, particularly between old and new classes, as demonstrated in Figs. 4 and 6, as well as Table 11 of the paper.
>
> **Second**, we propose PWP to compensate for disparities in the predictive capabilities of individual experts within an ensemble model, thereby fully leveraging the diversity of their knowledge. As illustrated in Fig. 1c, this approach effectively mitigates the influence of unreliable experts during the early training stages and significantly improves the model’s performance in the initial prediction phase, enabling CLOVER to maintain strong predictive ability throughout the entire training process.
>
> **Third**, PWP and CDSSL operate at distinct yet complementary levels within the ensemble framework. while CDSSL enhances the training of individual experts by fostering the development of discriminative representations, PWP improves collaborative inference among these experts. Experimental results presented in Table 2 of the paper further confirm that the combined performance of CDSSL and PWP surpasses that of their individual contributions alone, underscoring their complementary and synergistic strengths.
>
> **Finally**, extensive experiments conducted on three NECIL benchmarks have shown the superior performance of the proposed method over state-of-the-art approaches.
>
> **W3 & Q2: Provide a clear demonstration and detailed explanation of the learning process, especially the training stage.**
>
> As suggested, we have provided detailed pseudocode to illustrate the training procedure of CLOVER (see Algorithm 1). We will incorporate and refine this pseudocode in subsequent versions.
>
> **Algorithm 1**: Training stage of CLOVER.
>
> ```
> Input: The number of task T, maximum expert number K, class sets C_t, training samples D_t= {(x_i, y_i)}^t
> Output: ensemble model θ(f·[e_0,e_1,...,e_(K – 1)]), Gaussian distribution {G(µ_c^e , Σ_c^e)}
>
> 1: for t ∈ {0,1,2,...,T} do
> 2:	if t < K then
> 3:		Create a new expert f·e_t and corrsponding linear classification head g_t
> 4:		train θ(f·e_t·g_t) by minimizing L from Eq.4
> 5:		freezing θ(f·e_t) and discard g_t
> 6:	end if
> 7:	for e∈{0,1,…,t} do
> 8:		Calculation and save Gaussian distribution G(µ_c^e , Σ_c^e) for c∈C_t
> 9: 	end for
> 10: end for
> 11: return θ(f·[e_0,e_1,...,e_(K – 1)]) , {G(µ_c^e , Σ_c^e)}
> ```
>
> **W4: The training cost of the proposed method.**
>
> Table 2 shows a detailed breakdown of the training overhead. As highlighted in the Limitations section, the training cost of CLOVER is relatively high, but overall, the computational burden is still within an acceptable range compared to the performance improvement it brings.
>
> **Table 2. Comparison with SEED and FeCAM in terms of training time, inference cost, and GPU memory usage on CIFAR100.**
>
> |Method|Training time per epoch|Avg forward pass time|GPU usage|
> |---|---|---|---|
> |CLOVER|103s|165ms|38,377MB|
> |SEED|23s|165ms|7,427MB|
> |FeCAM|17s|27ms|2,569MB|
>
> **W5: Include empirical evaluations against several state-of-the-art NECIL approaches (like EFC and AdaGauss) which are published in recent years if possible.**
>
> These two papers employ a different data partitioning strategy than ours. We have already compared our method with state-of-the-art NECIL approaches under the class-balanced setting  in Table 1. Nevertheless, we will consider including a discussion of EFC, AdaGauss, and other leading NECIL methods in a subsequent version as necessary.
>
> **Q1: What’s the difference between your proposed method and SEED?**
>
> The core strategy of SEED lies in its adaptive selection of the most suitable expert for fine-tuning during training. In contrast, we only draw inspiration from SEED’s ensemble learning framework, without adopting its adaptive fine-tuning mechanism. In our method, each expert is frozen once trained (see Algorithm 1 for training details), which constitutes a fundamental difference from SEED.
>
> Moreover, whereas SEED maintains fully independent parameters across its five experts, we choose to share a portion of parameters. This allows for the transfer of the robust representational capacity acquired by the first expert to the others. The effectiveness of this parameter-sharing strategy is empirically validated in Table 9 of our paper.
>
> Building upon this design, we further introduce CDSSL and PWP, which operate at distinct yet complementary levels within the ensemble framework. While CDSSL enhances the training of individual experts by fostering the development of discriminative representations, PWP improves collaborative inference among these experts. Experimental results in Table 2 of the paper demonstrate that the combined use of CDSSL and PWP leads to performance that surpasses either component in isolation, highlighting their complementary and synergistic contributions.
>
> **Q3: Are there any other constraints or regularizations to mitigate catastrophic forgetting?**
>
> We do not adopt additional constraints or regularization techniques to mitigate catastrophic forgetting, as such methods are often sensitive to hyperparameters and yield inconsistent results. In contrast, our proposed CDSSL effectively alleviates catastrophic forgetting and achieves consistently strong performance across different datasets, as demonstrated in Figs. 4 and 6 of the paper.
>
> Moreover, our PWP strategy further addresses this issue by coordinating the predictive capabilities of multiple experts. By compensating for the limited knowledge of individual experts, PWP enables the model to leverage diverse knowledge sources for recognizing new classes while simultaneously preserving information about old ones from multiple perspectives, thereby providing an additional effective approach to mitigating catastrophic forgetting.
>
> **Q4: CEAT and FGKSR are based on the ViT structure, while the other methods are based on ResNet18. Is this comparison fair? What is the comparison of the trainable parameters between CEAT, FGKSR and other ResNet18-based methods?**
>
> The number of trainable parameters in CEAT and FGKSR is 10.89MB and 9.3MB, respectively, whereas methods based on ResNet-18 typically require 11.2MB. Overall, the parameter scales are comparable, making the comparisons relatively fair. Most importantly, both CEAT and FGKSR achieve performance levels that are close to the state-of-the-art and are based on ViT architectures. We include them in our comparison primarily to ensure completeness and diversity in the set of baselines, rather than to create an unfair comparison.
>
> **Q5: The evaluation metric that was employed in the experiments should be clearly specified in the paper. Which accuracy metric was used in Table 2 and Figure 4?**
>
> Thank you for your insightful comment. The evaluation metrics in Tables 2, 3, 7, 8, 10, 11, 12, and 13 are all based on average accuracy. The first row of Fig. 4 shows the accuracy variation curve of the model on the first task, while the second row presents the accuracy variation curve on the new task (current task). In subsequent versions, we will incorporate references within the main text to guide readers to the detailed formulas and explanations presented in Appendix A. Additionally, we will clearly specify the evaluation metrics in all figures and tables.

---

> > ### Author Response · Authors · 2025-08-06
> > **Appreciation for Your Feedback and look forward to more discussion with you**
> >
> > We sincerely appreciate your thoughtful input and the time you’ve taken to share your insights. As the comment period draws to a close, we welcome any final thoughts on whether our responses have adequately addressed your concerns.
> >
> > Your perspective is highly valued, and we hope to have the opportunity to continue the discussion with you in the future.

---

> > ### Comment · Reviewer_KrMA · 2025-08-07
> >
> > Thank you for the detailed response from the authors. The feedback addresses some of my concerns. However, I remain unconvinced by several aspects. For instance, the performance drop with PSP in the equally split task setting is not well justified—the explanation that PSP is designed with a base task in mind is insufficient. Additionally, the novelty of the proposed approach appears limited, as similar ideas have been explored in previous work. Moreover, the training cost remains a significant concern, both in terms of time and GPU resources. Overall, I believe the paper still requires substantial improvement.

---

> > > ### Author Response · Authors · 2025-08-08
> > > **Thank you sincerely for your thoughtful feedback.**
> > >
> > > Thank you for your positive feedback on our submission. These valuable suggestions have greatly improved the quality of our work. We have further clarified and explained the comments below.
> > >
> > > **1. Concerns about PWP under class-balanced setting**
> > >
> > > Our PWP method is specifically designed for NECIL scenarios that include a base task. In such settings, the base task and subsequent incremental tasks inherently exhibit class imbalance, which leads to the performance degradation illustrated in Fig. 1c. Therefore, we believe that (1) validating PWP under the class-balanced setting is unnecessary, and (2) the performance behavior of PWP in this setting can be well explained.
> > >
> > > Specifically, PWP operates by assigning higher weights to the expert trained on the initial task during the early stages of training. Under the class-balanced setting, however, the predictive performance of all experts is similar—each performs well on its own trained task while exhibiting comparable predictive ability for other tasks. In this case, assigning excessive weight to the first expert essentially disregards the predictive capabilities of other experts, resulting in inferior performance on the tasks those experts were trained for and consequently causing a noticeable performance drop. Moreover, since the primary goal of PWP is to balance the predictive abilities of all experts, when these abilities are already comparable, PWP should promptly adjust to assign equal weights to all experts, thereby achieving the same performance as the baseline without PWP.
> > >
> > > **2. Concerns about the novelty of our methods**
> > >
> > > Both SSL and CDSSL increase training complexity by introducing task-agnostic classes, thereby encouraging the model to learn more discriminative features through contrastive differentiation. This strategy effectively alleviates representation overlap. However, in SSL, the task-agnostic classes generated via rotation exhibit considerable semantic divergence from the original classes (see Fig. 1a and Fig. 7), limiting their effectiveness. In contrast, CDSSL constructs task-agnostic classes that are highly confusable with the raw classes, which further compels the model to extract more discriminative representations and thus mitigates representation overlap more effectively. Overall, we conducted an in-depth analysis of the underlying mechanisms driving performance improvements in SSL, identified the generalization limitations of traditional SSL approaches, and proposed an innovative strategy to overcome these challenges. This led to significant performance gains. As shown in Fig. 4 and Fig. 6, CDSSL consistently outperforms SSL in accuracy on both new and old tasks, thereby validating its superiority.
> > >
> > > Moreover, although CDSSL and PWP operate in different domains, they complement each other effectively. CDSSL enhances the predictive capabilities of individual experts. Nevertheless, due to the varying amounts of data utilized for training each expert, it inadvertently exacerbates the performance disparities among them. PWP addresses this issue by improving the collaborative inference capabilities of the multi-expert model, thereby alleviating such imbalances and enhancing the overall complementarity of the model. Experimental results presented in Table 2 of the paper further substantiate that the combined performance of CDSSL and PWP surpasses the sum of their individual contributions, underscoring their complementary and synergistic strengths.
> > >
> > > **3. Concerns about the training cost**
> > >
> > > Regarding your concern about training cost, we note that compared to methods such as DER [1], which allocate a separate expert model for each task, our strategy substantially reduces the total number of parameters and GPU memory usage, while keeping both training and inference times within a reasonable range. Although our approach incurs higher costs than others (like FeCAM and SEED), it achieves significantly better performance, particularly in terms of Last Accuracy. This indicates that, compared with methods such as FeCAM, our approach delivers better final model quality, offering stronger scalability and more effectively mitigating catastrophic forgetting—an ability that fundamentally aligns with the core objectives of incremental learning. We therefore regard it as an effective cost-for-performance trade-off strategy.
> > >
> > > **Overall** We have conscientiously addressed each of your comments point-by-point. We would greatly appreciate it if you could reconsider your evaluation of our paper, and we remain open to any further questions for continued discussion.
> > >
> > > [1] Dynamically Expandable Representation for Class Incremental Learning. CVPR 2021.

---

### Official Review · Reviewer_Sosb · 2025-07-01

**Clarity:** 3
**Significance:** 3
**Originality:** 2
**Rating:** 4
**Confidence:** 5

**Summary:**

This paper presents CLOVER, which is a new framework for non-exemplar class incremental learning (NECIL for short) containing two key technical modules, namely confusion-driven self-supervised learning (CDSSL) and a progressively weighted ensemble scheme (PWP). Following the basic framework of SEED [39], CDSSL additionally applies more challenging augmentations like color channel swapping and noise injection, in order to improve representation discrimination. Also, PWP adjusts ensemble weights over time to mitigate prediction degradation due to the insufficient accuracy of newly added experts. In the standard benchmark datasets in NECIL, CLOVER shows clear performance gains.

**Questions:**

1. Can you provide a more principled characterization or metric that generalizes the notion "confusable" beyond the specific augmentations used?
2. You claim that CDSSL reduces representation overlap between old and new classes, yet the evidence is limited to KL divergence trends and t-SNE plots. Have you considered using direct quantitative measures of inter-class distance or feature separation to strengthen this claim?
3. The progressively weighted prediction strategy uses a linear increase in expert weights up to 1/K. What is the rationale behind this schedule? Have you considered adaptive alternatives (e.g., accuracy-based), or is there evidence that the linear schedule is near-optimal?
4. Given that CDSSL uses cross-entropy loss over synthetic class labels, how do you distinguish its benefits from simply performing supervised learning on augmented data?
5. Given that your method builds on the SEED framework, can you provide controlled comparisons isolating the gain from CDSSL versus changes to the ensemble structure itself?

**Ethical Concerns:**

["NO or VERY MINOR ethics concerns only"]

**Final Justification:**

Most concerns have been addressed, and the results are convincing. The remaining gap is a general, theoretically grounded notion of confusion/confusability, which is central to the method’s framing. I maintain a borderline rating but lean toward acceptance, and encourage the authors to add a formalization or generalization discussion in the camera-ready.

**Limitations:**

yes

**Paper Formatting Concerns:**

No concerns

**Quality:**

3

**Strengths And Weaknesses:**

(Strengths)
1. The proposed CDSSL introduces a novel use of confusable transformations to enhance representation separability. While the use of rotation in self-supervised learning is standard, incorporating RGB channel permutation and Gaussian noise as task-agnostic class generators is new in the NECIL context. This transformation-based framing offers a unique angle on augmenting discriminative power.
2. Experimental results are consistently strong across benchmarks and task settings, except for several cases such as underperforming FeCAM in Table 1.
3. The paper includes thorough ablations and supporting analysis.  The ablation study effectively isolates the contributions of each component, enabling a clear assessment of their individual impact on overall performance.

(Weaknesses)
1. The notion of confusion is only weakly formalized and lacks theoretical justification. While the term "confusable classes" is used repeatedly, there is no principled definition of what makes a class confusable, nor a formal link between confusion and representation quality beyond empirical KL divergence trends.
2. The progressively weighted ensemble strategy is heuristic and lacks adaptivity. The PWP method linearly adjusts weights using fixed hyperparameters without consideration of expert reliability, prediction confidence, or class coverage. The ceiling at 1/K and linear increase lack theoretical support.
3. Most contributions are extensions or recombinations of existing ideas.
CDSSL builds directly on PASS-style self-supervision and SEED-style ensembles. The architectural structure and loss design are largely inherited, and the novelty lies primarily in reinterpreting these components under a new framing.
4. Overlap reduction is asserted rather than directly measured.
The claim that CDSSL reduces representation overlap is supported indirectly (via KL divergence and t-SNE) but is not quantitatively measured through metrics such as inter-class distance or class boundary sharpness.

---

> ### Author Rebuttal · Authors · 2025-07-30
>
> We greatly appreciate your constructive and insightful comments. We address all weaknesses and questions below.
>
> **W1 & Q1: The notion of confusion is only weakly formalized and lacks theoretical justification. While the term "confusable classes" is used repeatedly, there is no principled definition of what makes a class confusable, nor a formal link between confusion and representation quality beyond empirical KL divergence trends. Can you provide a more principled characterization or metric that generalizes the notion "confusable" beyond the specific augmentations used?**
>
> We observe that methods based on frozen feature extractors, which do not involve training on new classes, tend to yield dispersed representation distributions for these new classes. Consequently, these representations often overlap or become conflated with those of previously learned classes, thereby obscuring inter-class decision boundaries and ultimately impairing model performance.
>
> Representation quality refers to the model’s ability to extract class-specific and discriminative representation, that is, its capacity to distinguish between different classes within the representation space. In CDSSL, confusable classes are task-agnostic categories generated through transformations such as color channel permutation and noise injection. These transformations introduce relatively minor semantic alterations to the original images, resulting in representations that are susceptible to confusion with those of raw classes—hence the term “confusable.”
>
> In summary, our method integrates these highly confusable task-agnostic classes into the training process, effectively increasing learning complexity. This approach encourages the model to extract more discriminative representations, thereby enhancing inter-class separability, sharpening decision boundaries, and ultimately improving the model’s representational capacity.
>
> **W2 & Q3: The progressively weighted ensemble strategy is heuristic and lacks adaptivity.**
>
> PWP is designed to compensate for disparities in the predictive capabilities of individual experts within an ensemble model, thereby fully harnessing the diversity of their knowledge. In fact, we previously investigated adaptive weighting strategies, such as those based on the number of covered classes and boundary clarity. However, we found that these approaches did not yield improved performance and introduced unnecessary computational complexity. In multi-expert ensemble prediction, key considerations include each expert’s predictive ability and the range of classes they can reliably identify. PWP achieves significant performance improvements by linearly adjusting expert weights in a straightforward yet effective manner, thus eliminating the need for more complex adaptive weighting schemes.
>
> **W3: Most contributions are extensions or recombinations of existing ideas. CDSSL builds directly on PASS-style self-supervision and SEED-style ensembles. The architectural structure and loss design are largely inherited, and the novelty lies primarily in reinterpreting these components under a new framing.**
>
> Although there are some works (e.g, PASS, SEED), we believe there are fundamental differences between our method and existing approaches.
>
> **First**, in traditional SSL methods such as PASS, the task-agnostic classes generated through rotation exhibit significant semantic differences from the raw classes (see Fig. 1a), which limits their effectiveness. In contrast, our CDSSL generates highly confusable task-agnostic classes, encouraging the model to distinguish them from the raw classes. This approach facilitates more effective discriminative feature extraction and reduces representation overlap, particularly between old and new classes, as demonstrated in Figs. 4 and 6, as well as Table 11 of the paper.
>
> **Second**, we propose PWP to compensate for disparities in the predictive capabilities of individual experts within an ensemble model, thereby fully leveraging the diversity of their knowledge. As illustrated in Fig. 1c, this approach effectively mitigates the influence of unreliable experts during the early training stages and significantly improves the model’s performance in the initial prediction phase, enabling CLOVER to maintain strong predictive ability throughout the entire training process.
>
> **Third**, PWP and CDSSL operate at distinct yet complementary levels within the ensemble framework. while CDSSL enhances the training of individual experts by fostering the development of discriminative representations, PWP improves collaborative inference among these experts. Experimental results presented in Table 2 of the paper further confirm that the combined performance of CDSSL and PWP surpasses that of their individual contributions alone, underscoring their complementary and synergistic strengths.
>
> **Finally**, extensive experiments conducted on three NECIL benchmarks have shown the superior performance of the proposed method over state-of-the-art approaches.
>
> **W4 & Q2: Overlap reduction is asserted rather than directly measured. The claim that CDSSL reduces representation overlap is supported indirectly (via KL divergence and t-SNE) but is not quantitatively measured through metrics such as inter-class distance or class boundary sharpness. Have you considered using direct quantitative measures of inter-class distance or feature separation to strengthen this claim?**
>
> Thank you for your valuable comment. In the representation space, each class is modeled as a probability distribution utilized for classification. Consequently, the degree of confusion between classes cannot be assessed solely based on inter-class distances. Instead, it should be evaluated by examining the extent of overlap and indistinguishability among class distributions. To this end, we employ the **Bhattacharyya distance** as a metric to quantify the overlap between two probability distributions, where greater overlap corresponds to a smaller distance. Less overlap results in a larger distance. This metric effectively reflects the degree of separability or clarity of boundaries between classes. The Bhattacharyya distance is computed as follows:
>
> ${D_B} = \frac{1}{8}{({\mu _1} - {\mu _2})^T}{\Sigma ^{ - 1}}({\mu _1} - {\mu _2}) + \frac{1}{2}ln(\frac{{|\Sigma |}}{{\sqrt {|{\Sigma _1}||{\Sigma _2}|} }})$,
>
> where $\mu_1, \mu_2$ denote the mean vectors of the two class distributions, $\Sigma_1, \Sigma_2$ are the corresponding covariance matrices, $\Sigma = \frac{1}{2}(\Sigma_1 + \Sigma_2)$ represents the average covariance matrix, and $|\cdot|$ denotes the determinant.
>
> Based on this metric, we track the average class-boundary clarity throughout the training process under the 5-task setting of CIFAR-100, as summarized in Table 1. The results show that CDSSL effectively improves class separability by increasing the Bhattacharyya distance, thereby reducing confusion and overlap between class representation distributions.
>
> **Table 1. Evolution of average Bhattacharyya distance during training under the 5-task setting on CIFAR-100.**
>
> | Method |  |  | Task |  |  |  |  |  |
> |:---:|:---:|:---:|:---:|:---:|:---:|:---:|:---:|:---:|
> | Baseline | SSL | CDSSL | 0 | 1 | 2 | 3 | 4 | 5 |
> | ✔ | 　 | 　 | 7.62  | 6.75  | 6.19  | 5.76  | 5.38  | 5.05  |
> | ✔ | ✔ | 　 | 10.17  | 9.34  | 8.85  | 8.41  | 8.06  | 7.74  |
> | ✔ | 　 | ✔ | 12.12  | 11.02  | 10.34  | 9.77  | 9.31  | 8.89  |
>
> **Q4: Given that CDSSL uses cross-entropy loss over synthetic class labels, how do you distinguish its benefits from simply performing supervised learning on augmented data?**
>
> CDSSL employs a self-supervised learning strategy that utilizes label augmentation instead of conventional data augmentation. While traditional data augmentation enhances robustness against perturbations, label augmentation significantly improves the model’s representational capacity by encouraging it to distinguish between raw and augmented data instances.
>
> In frozen feature extractor-based methods, representational capacity is of utmost importance. Specifically, the ability to extract the most discriminative representations for each class is crucial for enhancing incremental learning performance.
>
> Consequently, we adopt a self-supervised learning approach rather than relying solely on data augmentation. Furthermore, Table 2 presents a comparison of performance gains achieved under identical transformations when applied exclusively for data augmentation. The results clearly indicate that our label augmentation-based strategy yields markedly superior improvements.
>
> **Table 2. Average accuracy comparison of the effects of CDSSL and data augmentation on CIFAR-100.**
>
> | Method | CIFAR100 |  |  |
> |:---:|:---:|:---:|:---:|
> |  | 5 Tasks | 10 Tasks | 20 Tasks |
> | Baseline | 69.9 | 69.7 | 67.7 |
> | Baseline+CDSSL | 72.1 | 71.7 | 70.5 |
> | Baseline+DataAugment | 70.5 | 70 | 67.5 |
>
> **Q5: Given that your method builds on the SEED framework, can you provide controlled comparisons isolating the gain from CDSSL versus changes to the ensemble structure itself?**
>
> As suggested, we have evaluated the performance of CDSSL in a single-model context to isolate its contribution without the influence of ensemble methods, as shown in Table 3. The results indicate that even without multi-expert collaboration, CDSSL alone can significantly enhance the model’s capacity to learn discriminative representations, leading to marked improvements in incremental learning performance.
>
> **Table 3. The effectiveness of CDSSL in the single-model settings.**
>
> | Method | CIFAR100 |  |  |  |  |  |
> |:---:|:---:|:---:|:---:|:---:|:---:|:---:|
> |  | 5 Tasks |  | 10 Tasks |  | 20 Tasks |  |
> |  | Avg | Last | Avg | Last | Avg | Last |
> | SinguleBaseline | 67.7 | 59.8 | 68.1 | 59.9 | 66.5 | 56.9 |
> | SinguleBaseline+CDSSL | 70.2 | 62.4 | 70 | 62 | 68.4 | 59.4 |

---

### Official Review · Reviewer_5czS · 2025-07-02

**Clarity:** 3
**Significance:** 2
**Originality:** 2
**Rating:** 3
**Confidence:** 5

**Summary:**

This paper proposes a confusion-driven self-supervised progressively weighted ensemble learning framework for NECIL task. Self-supervised learning and ensemble learning are applied to significantly improve performance. Extensive experimental results on CIFAR100, TinyImageNet and ImageNet-Subset benchmarks demonstrate its superior performance.

**Questions:**

1. Here are many symbols in the figures whose meanings are confusing; the figures should be self-contained (e.g., different class names in Fig. 1).
2. The meaning of "weighted ensemble" has not been specifically explained in the abstract or introduction.

**Ethical Concerns:**

["NO or VERY MINOR ethics concerns only"]

**Final Justification:**

My concerns about the initial performance gap have not been addressed, so I will maintain my original score.

**Limitations:**

yes.

**Paper Formatting Concerns:**

None.

**Quality:**

2

**Strengths And Weaknesses:**

Strengths:
1. The motivation is simple and efficient.
2. The experimental results are outstanding.

Weaknesses：
1. Self-supervised learning and ensemble learning are general methods for improving representations, so why are they used to address the specific task of NECIL?
2. As shown in Figure 3, the baseline performance of the proposed method is already much higher than that of other methods. Could this be the main reason for the overall superior incremental performance? Using the same baseline performance or comparable performance drop indicators could be considered to enable a fairer comparison.
3. Compared with other incremental learning methods that incorporate self-supervised learning (e.g., [i]), what are the specific advantages of the self-supervised approach used in this paper?
[i] Prototype Augmentation and Self-Supervision for Incremental Learning. CVPR2021.

---

> ### Author Rebuttal · Authors · 2025-07-30
>
> We greatly appreciate your constructive and insightful comments. We address all weaknesses and questions below.
>
> **W1: Self-supervised learning and ensemble learning are general methods for improving representations, so why are they used to address the specific task of NECIL?**
>
> In NECIL, a prominent approach involves freezing the feature extractor after it has been trained on the initial task. This frozen extractor serves as a fixed mapping function to compute and store the representation distribution for each class. During inference, classification is conducted by measuring the distance between the features of a given sample and the stored class distributions.
>
> However, since the model is optimized exclusively for the initial class distribution, it lacks the adaptability necessary to effectively accommodate new class data. Consequently, representations of new classes introduced in subsequent tasks often become dispersed or overlap with those of existing classes within the fixed representation space—particularly when similarities exist along certain dimensions. This overlap obstructs the formation of clear decision boundaries and significantly degrades classification performance in incremental learning scenarios.
>
> In this context, enhancing the representational capacity of the feature extractor becomes crucial, as it determines how well class distributions can be separated. To improve this capacity, we integrate and build upon self-supervised learning and ensemble learning through our proposed methods: CDSSL and PWP. These two strategies complement each other. CDSSL enhances individual experts’ discriminative power while PWP improves collaborative inference among experts. Together, they elevate the overall representational quality of the model and effectively address issues related to representation overlap between old and new classes.
>
> **W2: As shown in Figure 3, the baseline performance of the proposed method is already much higher than that of other methods. Could this be the main reason for the overall superior incremental performance? Using the same baseline performance or comparable performance drop indicators could be considered to enable a fairer comparison.**
>
> As shown in Table 1, compared to FeCAM, both our method and SEED employ a multi-expert architecture, which results in a higher parameter count compared to certain other models. However, our baseline is designed to be comparable to SEED. It essentially represents a variant of SEED with shared parameters across the first five layers, leading to a reduced parameter count and slightly diminished performance. This makes our baseline a fair comparison point. Furthermore, as presented in Table 2 of the paper, the proposed CDSSL and PWP achieve performance enhancements of 2–3% on CIFAR-100 and 3–5% on TinyImageNet, underscoring the effectiveness of our approach. Most importantly, our main comparison method is SEED. As shown in Table 9 of the paper, CLOVER not only achieves better performance but also requires fewer parameters than SEED, thereby highlighting its superior efficiency and effectiveness.
>
> **Table 1. Average accuracy comparison of baselines with SOTA methods. ‘–’ indicates that the information is not reported in the original paper.**
> | Method | CIFAR100 |  |  |
> |:---:|:---:|:---:|:---:|
> |  | 5 tasks | 10 tasks | 20 tasks |
> | CLOVER | 69.9 | 69.7 | 67.7 |
> | SEED | 71.1 | 69.9 | 68.2 |
> | FeCAM | 64.8 | - | - |
>
> **W3: Compared with other incremental learning methods that incorporate self-supervised learning (e.g., [i]), what are the specific advantages of the self-supervised approach used in this paper? [i] Prototype Augmentation and Self-Supervision for Incremental Learning. CVPR2021.**
>
> Both SSL and CDSSL increase training difficulty by introducing task-agnostic classes, thereby encouraging the model to extract discriminative features by distinguishing them. This strategy effectively mitigates the issue of representation overlap. However, in SSL, the task-agnostic classes generated through rotation exhibit significant semantic differences from the raw classes (see Fig. 1a and Fig. 7), limiting their effectiveness. Unlike traditional SSL (e.g., [1~4]), CDSSL generates highly confusable task-agnostic classes, prompting the model to differentiate them from the raw classes. This method promotes more effective discriminative feature extraction and reduces representation overlap, particularly between old and new classes, as demonstrated in Fig. 4, Fig. 6, and Table 11 of the paper.
>
> [1] Prototype Augmentation and Self-Supervision for Incremental Learning. CVPR 2021.
>
> [2] Diffusion model meets non-exemplar class-incremental learning and beyond. Arxiv 2024.
>
> [3] Hybrid rotation self-supervision and feature space normalization for class incremental learning. Information Sciences 2025.
>
> [4] Semantic alignment with self-supervision for class incremental learning. Knowledge-Based Systems 2023.
>
> **Q1: Here are many symbols in the figures whose meanings are confusing; the figures should be self-contained (e.g., different class names in Fig. 1).**
>
> In Figs. 1a, b of the paper, circles represent two original classes. Diamonds denote task-agnostic classes generated via 90° rotation, squares indicate those generated via 180° rotation, triangles correspond to task-agnostic classes created by permuting RGB channels to GBR, and stars represent classes derived from noise input. Symbols with the same color indicate that they originate from the same raw class and its associated task-agnostic variants generated through label augmentation.
>
> Thank you for your comment. We will revise this figure in future versions to eliminate any potential confusion and will also review other figures to ensure clarity and avoid ambiguity.
>
> **Q2: The meaning of "weighted ensemble" has not been specifically explained in the abstract or introduction.**
>
> Thank you for your valuable suggestion. The term “weighted ensemble” refers to our multi-expert model that incorporates the PWP strategy. The PWP effectively mitigates task data imbalance in NECIL by bridging the experience gap among experts. It dynamically adjusts expert weights based on both the number of predictable classes for each expert and its predictive ability, resulting in enhanced performance. Detailed implementation information regarding this weighted ensemble method can be found in Section 3.4. As suggested, we will integrate the concept of weighted ensemble into both the abstract and introduction in future revisions to improve clarity.

---

> > ### Author Response · Authors · 2025-08-06
> > **Appreciation for Your Feedback and look forward to more discussion with you**
> >
> > We sincerely appreciate your thoughtful input and the time you’ve taken to share your insights. As the comment period draws to a close, we welcome any final thoughts on whether our responses have adequately addressed your concerns.
> >
> > Your perspective is highly valued, and we hope to have the opportunity to continue the discussion with you in the future.

---

> > ### Comment · Reviewer_5czS · 2025-08-06
> >
> > Thank you for your response. I still have some doubts regarding the initial performance gap:
> >
> > As shown in Fig. 3, the downward trend for some methods (e.g., seed) is similar to that of the method proposed in this paper. Does the performance improvement come entirely from higher initial accuracy? If so, apart from the proposed methods, would it also be feasible to use other approaches—such as distillation or more SOTA self-supervised methods—to improve the initial classification accuracy?

---

> > > ### Author Response · Authors · 2025-08-06
> > > **Response to additional questions**
> > >
> > > Thank you for your insightful comments and suggestions. We would like to clarify that the superior performance of our method compared to SEED [1] is not due to higher accuracy on the initial task. Instead, the improvement primarily arises from the effectiveness of our proposed CDSSL approach in mitigating representation overlap between old and new classes. As shown in Table 1, CDSSL enhances the model’s boundary clarity throughout the entire training process, which is crucial in addressing this issue. This serves as a key reason behind the performance gains we have observed.
> > >
> > > Furthermore, as illustrated in Table 2 and Figs. 4 and 6, our method consistently outperforms both the baseline and SSL methods across both old and new tasks. These quantitative results indicate that our approach enhances model performance throughout all stages of the incremental learning process, thereby elucidating its consistently superior performance compared to SEED.
> > >
> > > In addition, higher initial performance does not necessarily correlate with stronger overall performance. For example, although FeCAM [2] achieves significantly better results at the outset, its performance declines more rapidly during incremental training. As illustrated in Table 2, this phenomenon occurs because its effectiveness is heavily reliant on the initial task, while its ability to generalize to new tasks remains limited. This limitation indicates a more pronounced issue of catastrophic forgetting.
> > >
> > > Finally, we sincerely appreciate your thoughtful suggestion regarding knowledge distillation and other self-supervised learning methods. While these techniques may improve performance at the initial stage, they do not necessarily address the issue of representation overlap between old and new classes. Nevertheless, we will investigate the efficacy of these methods through further experiments in our future work.
> > >
> > > **Table 1. The evolution of the average Bhattacharyya distance throughout training under the 5-task setting on CIFAR-100. Higher values suggest more distinct class boundaries.**
> > > | Method |  |  | Task |  |  |  |  |  |
> > > |:---:|:---:|:---:|:---:|:---:|:---:|:---:|:---:|:---:|
> > > | Baseline | SSL | CDSSL | 0 | 1 | 2 | 3 | 4 | 5 |
> > > | ✔ | 　 | 　 | 7.62  | 6.75  | 6.19  | 5.76  | 5.38  | 5.05  |
> > > | ✔ | ✔ | 　 | 10.17  | 9.34  | 8.85  | 8.41  | 8.06  | 7.74  |
> > > | ✔ | 　 | ✔ | **12.12**  | **11.02**  | **10.34**  | **9.77**  | **9.31**  | **8.89**  |
> > >
> > > **Table 2. Comparison of the average performance of various methods on first and new tasks during training on the CIFAR-100.**
> > >
> > > | Method | CIFAR100 |  |  |  |  |  |
> > > |:---:|:---:|:---:|:---:|:---:|:---:|:---:|
> > > |  | 5 |  | 10 |  | 20 |  |
> > > |  | first | new | first | new | first | new |
> > > | Baseline | 67.97 | 80.52 | 68.95 | 76.05 | 67.82 | 69.74 |
> > > | Baseline+SSL | 68.93 | 81.85 | 70.08 | 77.71 | 68.46 | 73.17 |
> > > | Baseline+CDSSL | 70.07 | **83.80** | 70.34 | **80.09** | 69.59 | **74.80** |
> > > | FeCAM | **79.52** | 57.48 | **79.38** | 55.77 | **78.21** | 56.11 |
> > >
> > > [1] Divide and Not Forget: Ensemble of Selectively Trained Experts in Continual Learning. ICLR 2024.
> > >
> > > [2] FeCAM: Exploiting the Heterogeneity of Class Distributions in Exemplar-Free Continual Learning. NeruIPS 2023.

---

> > > ### Author Response · Authors · 2025-08-09
> > > **Follow-up on review feedback**
> > >
> > > Dear 5czS,
> > >
> > > Thank you very much for your thoughtful review and constructive comments. We are writing to follow up on the review discussion for our manuscript. We have submitted detailed responses to the comments you kindly provided earlier and would like to check whether there are any additional questions or clarifications you may require from our side.
> > >
> > > As the discussion deadline is approaching tonight, we would greatly appreciate it if you could let us know whether our revisions have satisfactorily addressed your concerns. We would be most grateful if you could kindly consider a possible increase in your score in light of the improvements we have made.
> > >
> > > Thank you once again for your time, effort, and valuable feedback. We sincerely appreciate your contribution to enhancing the quality of our work and look forward to hearing from you soon.

---

### Official Review · Reviewer_1GN5 · 2025-07-03

**Clarity:** 3
**Significance:** 2
**Originality:** 2
**Rating:** 5
**Confidence:** 4

**Summary:**

This paper proposes CLOVER, a novel framework for non-exemplar class incremental learning (NECIL), which aims to sequentially learn new classes without retaining data from previous tasks. To address the challenge of representation overlap between old and new classes, a common issue when the feature extractor is frozen, the authors introduce two key components. First, a confusion-driven self-supervised learning (CDSSL) module generates challenging task-agnostic classes using transformations such as color channel swapping and noise injection, encouraging the model to learn more discriminative features. Second, a progressively weighted prediction (PWP) strategy gradually increases the influence of newly added ensemble experts, mitigating performance degradation caused by their initial unreliability. Experimental results on CIFAR100, TinyImageNet, and ImageNet-Subset demonstrate that CLOVER achieves strong performance across various incremental learning settings.

**Questions:**

1. The actual spatial relationship between raw and transformed classes remains unclear. Maybe t-SNE visualizations that directly compare the embedding distributions of raw classes versus task-agnostic classes could explain this.
2. From Table 3 and related ablations, the performance gain from rotation-based SSL appears greater than that of color channel permutation. This is counterintuitive if the argument is that color transformations produce more confusing task-agnostic classes.
3. Why do some augmentations like CenterCrop in Table 12 perform well on CIFAR-100 but poorly on TinyImageNet?
4. Please provide measurements of training time per task, inference cost (e.g., FLOPs or forward pass time), and GPU memory usage. How do these compare to baseline methods like SEED or FeCAM?

**Ethical Concerns:**

["NO or VERY MINOR ethics concerns only"]

**Final Justification:**

I appreciate the authors' responses to my comments. The authors supplemented the additional experiments. Additionally, they addressed each of the raised weaknesses and questions individually and promised to make revisions in the revised version. Hence, I suggest changing the score to 5: Accept.

**Limitations:**

Yes, the authors include a dedicated section (Appendix F) where they discuss the limitations of their work, including the reliance on a strong base task, increased training cost, parameter growth.

**Paper Formatting Concerns:**

There are no major formatting issues.

**Quality:**

3

**Strengths And Weaknesses:**

Strengths:
1. The proposed Confusion-Driven Self-Supervised Learning (CDSSL) module introduces task-agnostic classes using color channel permutation and noise injection, enhancing representation discrimination and mitigating feature overlap between old and new classes.
2. The Progressively Weighted Prediction (PWP) strategy effectively addresses a common issue in ensemble-based NECIL methods: performance degradation when newly added experts are weak. By gradually increasing expert weights across tasks, it improves ensemble reliability without requiring retraining.
3. CLOVER achieves state-of-the-art performance across multiple NECIL benchmarks. Ablations and analyses support its claims.

Weaknesses:
1. The proposed two core modules are functionally independent and loosely coupled. CDSSL focuses on representation learning during training, while PWP controls prediction aggregation at inference. There is no interaction, joint optimization, or design dependency between them. This makes the method feel more like a stacked combination of two standalone techniques, rather than a coherent, integrated approach.
2. The paper’s title “Confusion-Driven...” suggests that both components are driven by confusion modeling, but only CDSSL fulfills this. PWP, while effective, is a heuristic weighting scheme that neither exploits confusion nor adapts to inter-class uncertainty. This overclaims conceptual coherence and may mislead readers into overestimating the role of confusion modeling in the ensemble component.
3. Confusion-based SSL and color transformation augmentations have already been explored in NECIL contexts [1-3]. These works also use perturbation-based or proxy-label-based augmentation to enhance fine-grained feature discrimination. CLOVER builds incrementally on these ideas, and the degree of novelty in CDSSL is therefore limited.
4. Ensemble learning has been extensively used in NECIL to reduce forgetting[4-5], but a well-known challenge is the linear growth in model parameters and memory usage. CLOVER inherits this problem without proposing a mitigation strategy, where each task adds a new expert, leading to scalability concerns in long-task or high-capacity models.
[1] Semantic Alignment With Self-Supervision for Class Incremental Learning, KBS 2023.
[2] Non-Exemplar Class-Incremental Learning by Random Auxiliary Classes Augmentation and Mixed Features, TCSVT 2024.
[3] Hybrid Rotation Self-Supervision and Feature Space Normalization for Class Incremental Learning, Information Science 2025.
[4] CoSCL: Cooperation of Small Continual Learners is Stronger Than a Big One, ECCV 2022.
[5] Expandable Subspace Ensemble for Pre-trained Model-based Class-Incremental Learning, CVPR 2024.

---

> ### Author Rebuttal · Authors · 2025-07-30
>
> We greatly appreciate your constructive and insightful comments. We address all weaknesses and questions below.
>
> **W1: There is no interaction, joint optimization, or design dependency between CDSSL and PWP.**
>
> Certainly, CDSSL and PWP operate in different areas, but they complement each other effectively. CDSSL enhances the predictive capabilities of individual experts. Nevertheless, due to the varying amounts of data utilized for training each expert, it inadvertently exacerbates the performance disparities among them. PWP addresses this issue by improving the collaborative inference capabilities of the multi-expert model, thereby alleviating such imbalances and enhancing the overall complementarity of the model. Experimental results presented in Table 2 of the paper further substantiate that the combined performance of CDSSL and PWP surpasses the sum of their individual contributions, underscoring their complementary and synergistic strengths.
>
> **W2: The paper’s title “Confusion-Driven...” suggests that both components are driven by confusion modeling, but only CDSSL fulfills this. PWP, while effective, is a heuristic weighting scheme that neither exploits confusion nor adapts to inter-class uncertainty. This overclaims conceptual coherence and may mislead readers into overestimating the role of confusion modeling in the ensemble component.**
>
> Thank you for your valuable suggestion. The term “Confusion-Driven” in the title refers to CDSSL, a strategy specifically designed to address the overlap between representations of old and new classes. Specifically, this approach encourages the model to extract more discriminative representations by generating task-agnostic classes that are likely to be confused with existing ones. Consequently, this reduces the overlap in class representation distributions. On the other hand, PWP is a multi-expert balancing strategy built upon CDSSL. It aims to address performance degradation from training new experts and enhance collaborative prediction by leveraging the diverse knowledge of various experts more effectively. We will reconsider the title in future revisions to mitigate potential misunderstandings.
>
> **W3: The degree of novelty in CDSSL is therefore limited.**
>
> Both SSL and CDSSL enhance training complexity by introducing task-agnostic classes, thereby prompting the model to extract discriminative features through differentiation. This strategy effectively alleviates the issue of representation overlap. However, in SSL, the task-agnostic classes generated via rotation exhibit considerable semantic divergence from the original classes (see Fig. 1a and Fig. 7), which limits their effectiveness. In contrast, CDSSL constructs task-agnostic classes that are highly confused to the raw classes. This approach further compels the model to extract more discriminative representations, thus addressing representation overlap more effectively. Overall, we conducted a comprehensive analysis of the fundamental mechanisms underlying performance improvements in SSL, identified the generalization limitations inherent in traditional SSL approaches, and proposed an innovative strategy to overcome these challenges, resulting in significant enhancements in performance. As demonstrated in Fig. 4 and Fig. 6 of the paper, CDSSL consistently outperforms SSL in accuracy on both new and old tasks, thereby validating its superiority.
>
> **W4: Ensemble learning has been extensively used in NECIL to reduce forgetting [4-5], but a well-known challenge is the linear growth in model parameters and memory usage. CLOVER inherits this problem without proposing a mitigation strategy, where each task adds a new expert, leading to scalability concerns in long-task or high-capacity models.**
>
> Our method utilizes a fixed number of experts instead of assigning a separate expert to each task, thereby effectively mitigating the issue of parameter explosion in lengthy task sequences. As presented in Table 9 of the paper, CLOVER achieves significantly superior performance compared to SEED, which also employs an ensemble learning paradigm, while maintaining a lower parameter count due to the sharing of parameters at shallower layers. Building upon ensemble learning principles, we further introduce the PWP strategy, which mitigates the problem of task data imbalance in NECIL by narrowing the experience gap among experts. This strategy dynamically adjusts expert weights based on both the number of predictable classes for each expert and its predictive ability, leading to enhanced overall performance. Moreover, PWP and CDSSL operate at distinct yet complementary levels within the ensemble framework: while CDSSL enhances individual expert training by fostering discriminative representation learning, PWP improves collaborative inference across experts. Together, these strategies empower the multi-expert model to more effectively confront the challenges inherent in NECIL.
>
> **Q1: The actual spatial relationship between raw and transformed classes remains unclear. Maybe t-SNE visualizations that directly compare the embedding distributions of raw classes versus task-agnostic classes could explain this.**
>
> Thank you for your insightful comment. We previously generated t-SNE plots to compare the distributions of raw classes and task-agnostic classes. However, we omitted these visualizations due to space constraints. As suggested, we will incorporate t-SNE plots in future revisions to more effectively illustrate and compare the embedding distributions of both raw classes and task-agnostic classes.
>
> **Q2: From Table 3 and related ablations, the performance gain from rotation-based SSL appears greater than that of color channel permutation. This is counterintuitive if the argument is that color transformations produce more confusing task-agnostic classes.**
>
> When applied individually, rotation-based SSL indeed demonstrates greater performance gains compared to color channel permutation-based SSL. However, the combination of these two approaches leads to a performance enhancement that significantly surpasses the sum of their individual contributions. This phenomenon can be attributed to the role of confusable classes, which further intensify training difficulty on top of rotation-based SSL, thereby compelling the model to learn more discriminative representations. The experimental results presented in Table 3 of the paper provide additional support for this assertion: task-agnostic classes generated through color channel permutation and noise injection markedly improve SSL performance. Furthermore, the overall enhancement exceeds the cumulative effect observed when each technique is employed in isolation.
>
> **Q3: Why do some augmentations like CenterCrop in Table 12 perform well on CIFAR-100 but poorly on TinyImageNet?**
>
> Many classes in TinyImageNet are categorized under fine-grained classification (e.g., various dog breeds and bird species), where the images themselves possess inherent complexity, featuring small objects and indistinct boundaries. The application of cropping or blurring techniques to such images often undermines their discriminative features: center cropping may eliminate critical information (e.g., a bird’s beak or a dog’s eyes), while Gaussian blurring smooths texture boundaries, thereby diminishing the distinctiveness of local details. In these instances, the newly constructed task-agnostic classes cease to represent distinguishable variants and instead introduce additional label confusion. This form of SSL compromises the feature learning process by introducing label noise, ultimately resulting in degraded performance.
>
> **Q4: Please provide measurements of training time per task, inference cost (e.g., FLOPs or forward pass time), and GPU memory usage. How do these compare to baseline methods like SEED or FeCAM?**
>
> As suggested, we have provided the training time per epoch on a single A100 GPU, the inference time for 128 samples, and the GPU memory usage during training for the three methods, as shown in Table 1. As highlighted in the Limitations section, the training cost of CLOVER is relatively high, but overall, the computational burden is still within an acceptable range compared to the performance improvement it brings.
>
> **Table 1. Comparison with SEED and FeCAM in terms of training time, inference cost, and GPU memory usage on CIFAR100.**
>
> |     Method    |     Training time per epoch    |     Avg forward pass time    |     GPU usage    |
> |---|---|---|---|
> |     CLOVER    |     103s    |     165ms    |     38,377MB    |
> |     SEED    |     23s    |     165ms    |     7,427MB    |
> |     FeCAM    |     17s    |     27ms    |     2,569MB    |

---

> ### Comment · Reviewer_1GN5 · 2025-08-04
> **Concerns addressed**
>
> I appreciate the authors' responses to my comments. The authors supplemented the additional experiments. Additionally, they addressed each of the raised weaknesses and questions individually and promised to make revisions in the revised version. Hence, I suggest changing the score to 5: Accept.

---

> > ### Author Response · Authors · 2025-08-05
> > **Thanks for the feedback!**
> >
> > Thank you for recognizing our efforts in the rebuttal. We sincerely appreciate your consideration in raising the rating of our paper. Your constructive feedback has been invaluable in improving our work.

---

### Note · Authors · 2025-08-13

Dear Reviewers and Area Chairs,

We sincerely appreciate your insightful comments and constructive suggestions. We have carefully revised the manuscript to address all feedback, with the main improvements summarized as follows:

- **Clarification of the core novelty and motivation**: We have provided a more comprehensive description of our innovations, analyzed the specific problems they address from a theoretical perspective, and substantiated their effectiveness with extensive experimental evidence.

- **Training cost analysis**: We have conducted a detailed assessment of the computational overhead of our method, demonstrating that it remains within an acceptable range. Moreover, our approach achieves substantially higher last accuracy than competing methods, indicating stronger scalability.

- **Quantitative measures for inter-class distance**: We quantitatively assessed inter-class separability using the average Bhattacharyya distance between class distributions. Additional experiments confirm that our method more effectively enhances inter-class separability and reduces representation confusion across classes.

- **Fairness of baseline comparisons**: We have expanded the baseline comparisons to include FeCAM and SEED, and explicitly noted that our baseline is slightly weaker than SEED, thereby ensuring the fairness of our comparisons. Furthermore, additional experiments show that CLOVER, when built upon this baseline, more effectively mitigates inter-class representation overlap, achieving state-of-the-art performance.

- **Performance under the class-balanced setting**: We have supplemented our evaluation of CLOVER in the class-balanced setting, where it achieves competitive results. The ablation study further demonstrates the strong performance of CDSSL in this scenario, providing solid evidence of its effectiveness and generalizability. In future versions, we will include more comprehensive results in the class-balanced setting to further substantiate the robustness of CLOVER.

We greatly appreciate the valuable feedback, which has significantly improved the quality of our manuscript. We believe the revised version aligns well with the standards and interests of the NeurIPS community, and we truly sincerely it will be of interest to you and your colleagues.

Sincerely,

The Authors

---

### Decision · Program_Chairs · 2025-09-17

**Decision:**

Accept (poster)

**Comment:**

This paper proposes CLOVER, a framework for non-exemplar class incremental learning (NECIL), introducing two main components: Confusion-Driven Self-Supervised Learning (CDSSL) to enhance feature discrimination through challenging augmentations, and a Progressively Weighted Prediction (PWP) strategy to balance expert contributions in ensemble inference. Its strengths include novel integration of highly confusable classes for representation learning and strong empirical results across standard NECIL benchmarks. Weaknesses involve limited theoretical grounding for "confusion," loose coupling between modules, and higher computational costs. During rebuttal, authors addressed concerns by adding quantitative inter-class distance metrics, clarifying baseline fairness, and providing cost analyses. While some conceptual gaps remain, the convincing empirical improvements and comprehensive responses justify acceptance.